# Revealing Distribution Discrepancy by Sampling Transfer in Unlabeled Data

**Zhilin Zhao**[1,2]     **Longbing Cao**[1]     **Xuhui Fan**[1]     **Wei-Shi Zheng**[2,3*]

[1] School of Computing, Macquarie University, Australia
[2] School of Computer Science and Engineering, Sun Yat-sen University, China
[3] Key Laboratory of Machine Intelligence and Advanced Computing, Ministry of Education, China
zhaozhl7@hotmail.com, {longbing.cao,xuhui.fan}@mq.edu.au, wszheng@ieee.org

## Abstract

There are increasing cases where the class labels of test samples are unavailable, creating a significant need and challenge in measuring the discrepancy between training and test distributions. This distribution discrepancy complicates the assessment of whether the hypothesis selected by an algorithm on training samples remains applicable to test samples. We present a novel approach called Importance Divergence (I-Div) to address the challenge of test label unavailability, enabling distribution discrepancy evaluation using only training samples. I-Div transfers the sampling patterns from the test distribution to the training distribution by estimating density and likelihood ratios. Specifically, the density ratio, informed by the selected hypothesis, is obtained by minimizing the Kullback-Leibler divergence between the actual and estimated input distributions. Simultaneously, the likelihood ratio is adjusted according to the density ratio by reducing the generalization error of the distribution discrepancy as transformed through the two ratios. Experimentally, I-Div accurately quantifies the distribution discrepancy, as evidenced by a wide range of complex data scenarios and tasks.

## 1 Introduction

The assumption that data are independently and identically distributed (IID) is staple in statistical machine learning. It suggests that a hypothesis selected by an algorithm, after observing several training samples, should perform effectively on test samples from the same unknown distribution. However, this assumption often oversimplifies the intricate and diverse nature of real-world data, particularly in non-IID scenarios [1, 2]. Thus, if training samples are considered in-distribution (ID), there is a risk that test samples may deviate from this distribution, characterized as out-of-distribution (OOD) [3]. This distribution discrepancy between training and test distributions poses a critical and challenging non-IID learning [2] question: *How to quantify the applicability of a hypothesis derived from training samples to test samples?*. This question is generally applicable to areas like OOD generalization [4], OOD detection [5, 6], domain adaptation [7, 8], transfer learning [9], semi-supervised learning [10], robust learning [11], and adversarial learning [12].

The applicability of a hypothesis can be determined by the distribution discrepancy between training and test distributions. When the two distributions align, meaning both training and test samples are ID, the hypothesis derived from training samples can be seamlessly applied to test samples. However, in reality, particularly when test samples fall OOD, this scenario rarely holds. Under such circumstances, decisions must be based on the extent of the distribution discrepancy. This may require enhancing the hypothesis generalization capability towards test samples or rejecting these

---

[*]Corresponding author

samples outright. These actions are fundamental to the principles of OOD generalization and OOD detection, respectively [13].

Evaluating the distribution discrepancy between training and test distributions for a selected hypothesis presents a significant challenge, as training samples are typically labeled, whereas test samples often are not [14]. This discrepancy means that conventional statistical distances, such as F-divergences [15], integral probability metrics [16], and total variation [17], are not suitable for this situation. Furthermore, density ratio methods [18, 19] offer a partial solution by disregarding label information and estimating the ratio between input distributions. Moreover, more performance prediction techniques [20] aim to navigate this challenge by examining the uncertain outcomes of the hypothesis, potentially leading to unreliable results. A detailed discussion of related work is provided in Appendix A.

To tackle the issue of unavailable test labels, we introduce the concept of *Importance Divergence* (I-Div), which measures the training-test distribution discrepancy w.r.t. the difference between the expected risks of the selected hypothesis on training and test distributions. To estimate the expected risk on test distributions without label access, the core strategy involves importance sampling to transfer the sampling patterns from the test distribution to the training distribution. This process requires the estimation of density and likelihood ratios. Specifically, the density ratio, informed by the selected hypothesis, is obtained by minimizing the Kullback-Leibler divergence between the actual and estimated input distributions. Simultaneously, the likelihood ratio is adjusted according to the density ratio by reducing the generalization error related to the distribution discrepancy as transformed through the two ratios. As a result, I-Div leverages the estimated density and likelihood ratios to quantitatively measure the distribution discrepancy between training and test distributions, eliminating the need for test class labels, and thus quantifying the applicability of the hypothesis across different datasets.

## 2  Preliminaries

Let $\mathcal{X}$ denote the input space, and $\mathcal{Y}$ represent the set of labels. The joint probability distributions are represented by $\mathcal{P}$ for training samples and $\mathcal{Q}$ for test samples. Assume we observe a labeled training dataset $\widehat{\mathcal{P}}$ and a unlabeled test dataset $\widehat{\mathcal{Q}}$ containing $N$ IID samples from $\mathcal{P}$ and $\mathcal{Q}$, respectively. $\mathcal{Q}_{\mathcal{X}}$ denotes the marginal distribution of $\mathcal{Q}$ over $\mathcal{X}$. $\widehat{\mathcal{P}}$ and $\widehat{\mathcal{Q}}$ are described as

$$\widehat{\mathcal{P}} = \{(\mathbf{x}_i, y_i)\}_{i=1}^{N} \overset{\text{IID}}{\sim} \mathcal{P}, \quad \widehat{\mathcal{Q}} = \{\mathbf{x}_i\}_{i=1}^{N} \overset{\text{IID}}{\sim} \mathcal{Q}_{\mathcal{X}}, \tag{1}$$

where the label space $\mathcal{Y}$ encompasses all labels of samples from both $\mathcal{P}$ and $\mathcal{Q}$. We also define $\mathcal{H}$ as the hypothesis space and $\mathfrak{L}(\cdot, \cdot) \in (0, B_{\mathcal{L}})$ as the bounded loss function. The expected and empirical risks [21] for a hypothesis $h \in \mathcal{H} : \mathcal{X} \to \mathcal{Y}$ on distribution $\mathcal{P}$ and the dataset $\widehat{\mathcal{P}}$ are defined as $\epsilon_{\mathcal{P}}(h)$ and $\widehat{\epsilon}_{\widehat{\mathcal{P}}}(h)$, respectively, i.e.,

$$\epsilon_{\mathcal{P}}(h) = \mathbb{E}_{\mathcal{P}}[\mathfrak{L}(h(\mathbf{x}), y)] = \int_{\mathcal{X}} \int_{\mathcal{Y}} \mathfrak{L}(h(\mathbf{x}), y) \, \mathcal{P}(\mathbf{x}, y) \, d\mathbf{x} \, dy,$$

$$\widehat{\epsilon}_{\widehat{\mathcal{P}}}(h) = \widehat{\mathbb{E}}_{\widehat{\mathcal{P}}}[\mathfrak{L}(h(\mathbf{x}), y)] = \frac{1}{|\widehat{\mathcal{P}}|} \sum_{(\mathbf{x}, y) \in \widehat{\mathcal{P}}} \mathfrak{L}(h(\mathbf{x}), y), \tag{2}$$

where $\mathbb{E}$ and $\widehat{\mathbb{E}}$ represent the expectation with respect to a data distribution and the sample average over a dataset, respectively. Accordingly, an algorithm $\mathcal{A}$ aims to select the empirical risk minimizer $\widehat{h}_{\widehat{\mathcal{P}}}$ after observing the samples from the training dataset $\widehat{\mathcal{P}}$ by

$$\widehat{h}_{\widehat{\mathcal{P}}} \in \arg \min_{h \in \mathcal{H}} \widehat{\epsilon}_{\widehat{\mathcal{P}}}(h), \tag{3}$$

to approximate the optimal hypothesis $h_{\mathcal{P}}^*$ selected from the distribution $\mathcal{P}$ through $h_{\mathcal{P}}^* \in \arg \min_{h \in \mathcal{H}} \epsilon_{\mathcal{P}}(h)$.

Our central research question is formulated as follows: *How can we quantify the applicability of the minimizer $\widehat{h}_{\widehat{\mathcal{P}}}$, originated from the training samples $\widehat{\mathcal{P}}$, to the unlabeled test samples $\widehat{\mathcal{Q}}$?* To address this question quantitatively, we delve into methodologies for assessing the distribution discrepancy between training and test distributions, particularly focusing on doing so without the need to access class labels from test samples.

## 3 Importance divergence

For a hypothesis chosen by an algorithm after observing training samples, to assess its applicability to test samples in the absence of ground truth labels, we introduce the concept of Importance Divergence (I-Div). I-Div estimates the distribution discrepancy between training and test distributions for the given hypothesis. To address the issue of unavailable ground truth labels, it leverages importance sampling, density ratios, and likelihood ratios, facilitating the sampling transfer in the test distribution back to the training distribution.

In this section, we first present the distribution discrepancy with importance sampling, which transfers the data sampling from the test distribution to the training distribution. Following this, we discuss the methodologies for estimating the hypothesis-oriented density and adaptive likelihood ratios, essential components of this discrepancy, to facilitate this sampling transfer. The hypothesis-oriented density ratio is specifically tailored to each hypothesis, as it assesses the suitability of a specific hypothesis based on the resulting distribution discrepancy. The adaptive likelihood ratio is adjusted according to the density ratio to expedite the convergence of the distribution discrepancy. Lastly, we utilize an empirical estimator of the distribution discrepancy to evaluate the applicability of a hypothesis selected from training samples to test samples.

### 3.1 Distribution discrepancy with importance sampling

For the hypothesis $\widehat{h}_{\widehat{\mathcal{P}}}$, I-Div evaluates the distribution discrepancy between training and test distributions without requiring the ground truth labels of the test samples. A smaller discrepancy implies that the training and test samples could be considered as drawn from the same distribution with respect to the given hypothesis, and vice versa. By using variational divergence, we can express this discrepancy as the difference between the expected risks of the hypothesis on training and test distributions by

$$d(\mathcal{P}, \mathcal{Q} \mid \widehat{h}_{\widehat{\mathcal{P}}}) = \left| \epsilon_{\mathcal{P}}(\widehat{h}_{\widehat{\mathcal{P}}}) - \epsilon_{\mathcal{Q}}(\widehat{h}_{\widehat{\mathcal{P}}}) \right|. \tag{4}$$

If the training and test distributions are aligned, i.e., $\mathcal{P} = \mathcal{Q}$, then the expected risks for the hypothesis $\widehat{h}_{\widehat{\mathcal{P}}}$ are similar, resulting in a minimal distribution discrepancy. This condition suggests that the test samples are likely ID for the hypothesis $\widehat{h}_{\widehat{\mathcal{P}}}$. Conversely, a notable difference between $\mathcal{P}$ and $\mathcal{Q}$ indicates a larger distribution discrepancy, implying that the hypothesis $\widehat{h}_{\widehat{\mathcal{P}}}$ perceives the training and test samples as originated from distinct distributions, thus categorizing the test samples as OOD.

To make the expected risk difference in Eq. (4) more pronounced, I-Div employs Jensen's inequality to consider its upper bound by

$$d(\mathcal{P}, \mathcal{Q} \mid \widehat{h}_{\widehat{\mathcal{P}}}) \leq \int_{\mathcal{Y}} \int_{\mathcal{X}} |\mathcal{P}(\mathbf{x}, y) - \mathcal{Q}(\mathbf{x}, y)| \, \mathfrak{L}\left(\widehat{h}_{\widehat{\mathcal{P}}}(\mathbf{x}), y\right) d\mathbf{x} \, dy, \tag{5}$$

thereby more distinctly highlighting the differences between the training and test distributions.

Recall that class labels in test samples are inaccessible, thus, the principal challenge is evaluating the expected risk for the test distribution with respective to the given hypothesis without access to ground truth labels. Since direct sampling from the test distribution is not feasible, an alternative is sampling from the training distribution. To overcome this limitation, we employ the importance sampling technique [22], converting the data sampling from test to training distributions, i.e.,

$$\mathcal{Q}(\mathbf{x}, y) = \frac{\mathcal{Q}(\mathbf{x}, y)}{\mathcal{P}(\mathbf{x}, y)} \cdot \mathcal{P}(\mathbf{x}, y) = \underbrace{\frac{\mathcal{Q}(\mathbf{x})}{\mathcal{P}(\mathbf{x})}}_{r(\mathbf{x})} \cdot \underbrace{\frac{\mathcal{Q}(y \mid \mathbf{x})}{\mathcal{P}(y \mid \mathbf{x})}}_{v(\mathbf{x}, y)} \cdot \mathcal{P}(\mathbf{x}, y), \tag{6}$$

where $r$ and $v$ denote a density ratio and a likelihood ratio, respectively. Thus, I-Div estimates distribution discrepancy between training and test distributions by merely sampling from the training one without accessing the class labels of the test samples. According to Eq. (5) and Eq. (6), we have

$$d\left(\mathcal{P}, \mathcal{Q} \mid \widehat{h}_{\widehat{\mathcal{P}}}, r, v\right) \triangleq \mathbb{E}_{\mathcal{P}}\left[|r(\mathbf{x})v(\mathbf{x}, y) - 1| \, \mathfrak{L}\left(\widehat{h}_{\widehat{\mathcal{P}}}(\mathbf{x}), y\right)\right]. \tag{7}$$

Accordingly, for a given labeled training dataset $\widehat{P}$ and a unlabeled test dataset $\widehat{Q}$, we can then construct an empirical estimator for estimating the distribution discrepancy in Eq. (7) by

$$\widehat{d}\left(\widehat{\mathcal{P}}, \widehat{\mathcal{Q}} \mid \widehat{h}_{\widehat{\mathcal{P}}}, r, v\right) \triangleq \widehat{\mathbb{E}}_{\widehat{\mathcal{P}}}\left[|r(\mathbf{x})v(\mathbf{x}, y) - 1| \, \mathfrak{L}(\widehat{h}_{\widehat{\mathcal{P}}}(\mathbf{x}), y)\right]. \tag{8}$$

However, to estimate the distribution discrepancy I-Div, the emerging challenge involves determining the density ratio $r$ and likelihood ratio $v$ from observed samples, which are discussed in Section 3.2 and Section 3.3, respectively.

## 3.2 Hypothesis-oriented density ratio

The density ratio in the distribution discrepancy should be hypothesis-oriented. That is, it should depend on the specific hypothesis $\widehat{h}_{\widehat{\mathcal{P}}} \in \mathcal{H}$ selected by an algorithm $\mathcal{A}$ on $\widehat{\mathcal{P}}$. This is because the criteria for judging the discrepancy between distributions $\mathcal{P}$ and $\mathcal{Q}$ vary across different algorithms. For instance, whether two datasets of cats and dogs respectively come from the same distribution depends on whether the algorithm aims to identify if the subjects are biological entities or to distinguish between these two species. Accordingly, we apply a deep neural network to model a density ratio $r$ from the hypothesis space $\mathcal{R}(\widehat{h}_{\widehat{\mathcal{P}}})$ depending on the specific hypothesis $\widehat{h}_{\widehat{\mathcal{P}}}$. Then, we select a density ratio $r \in \mathcal{R}(\widehat{h}_{\widehat{\mathcal{P}}})$ to minimize the Kullback-Leibler divergence between the actual distribution and the estimated distribution based on this density ratio.

To construct the hypothesis space $\mathcal{R}(\widehat{h}_{\widehat{\mathcal{P}}})$, we utilize the output representations from $\widehat{h}_{\widehat{\mathcal{P}}}$ with a learnable component $\omega$ to model a density ratio $r$. Specifically, we decompose the hypothesis into a backbone $\psi$ and a softmax layer $\phi$, represented as $\widehat{h}_{\widehat{\mathcal{P}}}(\mathbf{x}) = (\phi \circ \psi)(\mathbf{x})$, where the output dimension of the backbone is $O_\psi$. Using a learnable component $\omega$, $r$ is constructed as $r(\mathbf{x}) = (\omega \circ \psi)(\mathbf{x})$, where the learnable component $\omega(\cdot) = (\omega_{\mathrm{SP}} \circ \omega_{\mathrm{AD}})(\cdot)$ contains an adapter $\omega_{\mathrm{AD}}$ [23] to introduce learnable parameters and a Softplus layer $\omega_{\mathrm{SP}}$ [24] to ensure strictly positive outputs. Specifically, the adapter $\omega_{\mathrm{AD}}$, which follows the bottleneck $\psi$, comprises two fully connected layers with a Gaussian Error Linear Units (GELU) [25] activation layer in between. Furthermore, the Softplus layer is adopted to the adapter output, effectively mapping it to the range $(0, +\infty]$. Furthermore, weight matrices are $\mathbf{W}_1 \in \mathcal{W}_1 \in \mathbb{R}^{O_\psi \times O_m}$ and $\mathbf{W}_2 \in \mathcal{W}_2 \in \mathbb{R}^{O_m \times O_\omega}$ in the fully connected layers. The activation functions GELU and Softplus are $\beta_1$- and $\beta_2$-Lipschitz, respectively. Additionally, we assume $\tau_1 = \sup_{\mathbf{W}_1 \in \mathcal{V}_1} \|\mathbf{W}_1\|_{1,\infty}$ and $\tau_2 = \sup_{\mathbf{W}_2 \in \mathcal{V}_2} \|\mathbf{W}_2\|_{1,\infty}$. Thus, the density ratio $r \in \mathcal{R}(\widehat{h}_{\widehat{\mathcal{P}}})$ can be modeled as

$$r(\mathbf{x}) = (\omega_{\mathrm{SP}} \circ \omega_{\mathrm{AD}} \circ \psi)(\mathbf{x}) = \ln\left(1 + \exp\left(\frac{1}{|O_\omega|} \sum_{i \in [O_\omega]} (\omega_{\mathrm{AD}} \circ \psi)(\mathbf{x})_i\right)\right), \qquad (9)$$

where $O_\omega$ represents the output dimensionality of the adapter. Without loss of generality, we further assume that $r(\mathbf{x}) \in (b_r, B_r)$ for any $r \in \mathcal{R}(\widehat{h}_{\widehat{\mathcal{P}}})$ and $\mathbf{x} \in \mathcal{X}$.

To select a density ratio $r \in \mathcal{R}(\widehat{h}_{\widehat{\mathcal{P}}})$, we use it to estimate the density $\mathcal{P}$ and $\mathcal{Q}$ by

$$\widetilde{\mathcal{P}}(\mathbf{x}) = \mathcal{Q}(\mathbf{x})/r(\mathbf{x}), \quad \widetilde{\mathcal{Q}}(\mathbf{x}) = \mathcal{P}(\mathbf{x}) \cdot r(\mathbf{x}). \qquad (10)$$

Drawing on the inspiration of importance estimation methods [26, 27], we construct objectives and constraints around $\widetilde{\mathcal{P}}(\mathbf{x})$ and $\widetilde{\mathcal{Q}}(\mathbf{x})$. The estimated probability distributions are designed to approximate their actual counterparts, suggesting the minimization of the two KL divergences with normalization constraints

$$\min_{r \in \mathcal{R}(\widehat{h}_{\widehat{\mathcal{P}}})} \quad \mathrm{KL}\left(\mathcal{P}(\mathbf{x}) \| \widetilde{\mathcal{P}}(\mathbf{x})\right) + \mathrm{KL}\left(\mathcal{Q}(\mathbf{x}) \| \widetilde{\mathcal{Q}}(\mathbf{x})\right),$$
$$\text{s.t.} \quad \int \widetilde{\mathcal{P}}(\mathbf{x})\,d\mathbf{x} = 1, \int \widetilde{\mathcal{Q}}(\mathbf{x})\,d\mathbf{x} = 1. \qquad (11)$$

For convenience, we assume $\mathcal{U} = \mathcal{P}/2 + \mathcal{Q}/2$ and define $\mathcal{C} = \{1, -1\}$ as labels for training and test samples, respectively. A label $c \in \mathcal{C}$, corresponding to a sample from $\mathcal{U}$, indicates its distribution origin. The assignment $c = 1$ indicates that a sample originates from distribution $\mathcal{P}$, while $c = -1$ signifies that a sample comes from distribution $\mathcal{Q}$. We can then obtain the following objective function for learning the density ratio

$$f(r) = \mathbb{E}_{(\mathbf{x},c) \sim \mathcal{U}}\left[c \log r(\mathbf{x}) + \lambda \|(r(\mathbf{x}))^c - 1\|^2\right], \qquad (12)$$

where $\lambda \geq 0$ balances the KL divergence and normalization constraints, as the detailed derivation shown in Appendix B. We further assume that $f(r)$ is $L_f$-lipschitz continuous with respect to

$r \in \mathcal{R}(\widehat{h}_{\widehat{\mathcal{P}}})$. This objective function Eq. (12) can be estimated by

$$\widehat{f}(r) = \widehat{\mathbb{E}}_{(\mathbf{x},c)\sim\widehat{\mathcal{U}}} \left[ c \log r(\mathbf{x}) + \lambda \left\| (r(\mathbf{x}))^c - 1 \right\|^2 \right], \tag{13}$$

where $\widehat{\mathcal{U}} = \widehat{\mathcal{P}} \cup \widehat{\mathcal{Q}}$. An empirical risk minimizer $\widehat{r}$ is selected by

$$\widehat{r} \in \arg \min_{r\in\mathcal{R}(\widehat{h}_{\widehat{\mathcal{P}}})} \widehat{f}(r), \tag{14}$$

which aims to approximate the population risk minimizer $\bar{r} \in \arg\min_{r\in\mathcal{R}(\widehat{h}_{\widehat{\mathcal{P}}})} f(r)$. The convergence rate of $\widehat{r}$ can be guaranteed by the following theorem.

**Theorem 3.1.** *Let $\lambda \geq 1$ and $\mu$ be a constant related to the function $f$. With a probability of at least $1 - \delta$,*

$$\mathbb{E}_{\mathcal{P}} \left| \bar{r}(\mathbf{x}) - \widehat{r}(\mathbf{x}) \right|^2 \leq \frac{64 L_f (\beta_1\beta_2\tau_1\tau_2\sqrt{O_\psi + 1} + B_r\sqrt{\ln 4/\delta})}{\mu\sqrt{N}} + \frac{8\beta_2\tau_2 L_f}{\mu N} := \mathfrak{B}(\delta, N).$$

The presence of $N$ in the denominators of both terms suggests that the bound tightens with an increasing sample size, which aligns with the general understanding that more data can lead to more accurate estimates in statistical learning.

### 3.3 Adaptive likelihood ratio

Without making further assumptions, it is infeasible to estimate the likelihood ratio $v$ due to its dependence on the unknown joint distribution $\mathcal{Q}(\mathbf{x}, y)$. Instead of pursuing the true likelihood ratio, our goal is to approximate an adaptive likelihood ratio $v \in \mathcal{V}(\widehat{r})$ that enables a swift convergence of the distribution discrepancy, guided by the hypothesis-oriented density ratio $\widehat{r} \in \mathcal{R}(\widehat{h}_{\widehat{\mathcal{P}}})$. This strategy is valid since the density ratio captures the input distribution discrepancy between training and test distributions. It indicates that utilizing even a basic form of the covariate shift assumption, i.e., $v(\mathbf{x}, y) = 1$ for all $(\mathbf{x}, y) \sim \mathcal{P}$, allows the distribution discrepancy, as calculated by Eq. (8), to approximate the difference between distributions to a reasonable degree. Furthermore, since the density ratio serves primarily to gauge the distribution discrepancy, the corresponding likelihood ratio must be specifically adapted to this density ratio for precisely assessing the distribution discrepancy.

Accordingly, we reveal the generalization error bound of the distribution discrepancy, leveraging both density and likelihood ratios, through the convergence rate of the hypothesis-oriented density ratio, Rademacher complexity [28] and Talagrand's contraction lemmas [29], which is shown as follows.

**Theorem 3.2.** *Based on the conditions and results outlined in Theorem 3.1, with a probability of at least $1 - \delta$, $\left| d\left( \mathcal{P}, \mathcal{Q} \mid \widehat{h}_{\widehat{\mathcal{P}}}, \bar{r}, v \right) - \widehat{d}\left( \widehat{\mathcal{P}}, \widehat{\mathcal{Q}} \mid \widehat{h}_{\widehat{\mathcal{P}}}, \widehat{r}, v \right) \right|$ is bounded by*

$$B_{\mathfrak{L}} \sqrt{\frac{\ln(2/\delta)\sum_{(\mathbf{x},y)\sim\widehat{\mathcal{P}}} |\widehat{r}(\mathbf{x})v(\mathbf{x}, y) - 1|^2}{N}} + B_{\mathfrak{L}} \mathbb{E}_{\mathcal{P}}\left[ v(\mathbf{x}, y) \right] \sqrt{\frac{\mathfrak{B}(\delta/2, N)}{\mu}}.$$

The result shows that the estimated distribution discrepancy converges quickly with increasing the sample size. Additionally, the bound is associated with the values of the likelihood ratio. Moreover, as per Eq. (6), for any $r \in \mathcal{R}(\widehat{h}_{\widehat{\mathcal{P}}})$, we have

$$\int\int \mathcal{Q}(\mathbf{x}, y)\, d\mathbf{x}\, dy = \mathbb{E}_{\mathcal{P}}[r(\mathbf{x})v(\mathbf{x}, y)] = 1. \tag{15}$$

Considering the terms in Theorem 3.2 related to $v$ and the average error over the samples from $\widehat{P}$, we have

$$\min_{v\in\mathcal{V}(\widehat{r})} \widehat{\mathbb{E}}_{\widehat{\mathcal{P}}}\left[ v(\mathbf{x}, y) + \frac{\gamma}{2}(\widehat{r}(\mathbf{x})v(\mathbf{x}, y) - 1)^2 \right],$$

$$\text{s.t.} \quad \widehat{\mathbb{E}}_{\widehat{\mathcal{P}}}[\widehat{r}(\mathbf{x})v(\mathbf{x}, y)] = 1, \tag{16}$$

where $\gamma > 0$ acts as a regularization parameter, influencing the trade-off, and the optimal solution is $\bar{v}$. By using an proximal algorithm [30], we can obtain the following approximate solution

$$\widehat{v}(\mathbf{x}, y) = \frac{N(\gamma\widehat{r}(\mathbf{x}) - 1)}{\gamma(\widehat{r}(\mathbf{x}))^2} \cdot \widehat{\mathbb{E}}_{\widetilde{\mathbf{x}}\in\widehat{\mathcal{P}}}\left[ \frac{\gamma(\widehat{r}(\widetilde{\mathbf{x}}))^2}{\gamma\widehat{r}(\widetilde{\mathbf{x}}) - 1} \right]. \tag{17}$$

We observe that in the two boundary cases where $\gamma \to 0$ and $\gamma \to +\infty$, $\widehat{v}$ consistently equals 1, thus adhering to the covariate shift assumption. Even under these extreme conditions, it is feasible to calculate the distribution discrepancy using both the hypothesis-oriented density ratio and adaptive likelihood ratio to overcome the challenge of unlabelled test samples. This approach is viable because the hypothesis-oriented density ratio $\widehat{r}$ quantifies the distribution differences without relying on class labels, while employing $\gamma \in (0, +\infty)$ utilizes the class labels of training samples. Although this method may not precisely determine the likelihood ratio for each instance, it is designed in accordance with a hypothesis-oriented density ratio such that the estimated distribution discrepancy aligns with the actual value, fulfilling our primary objective.

### 3.4 Hypothesis applicability evaluation

I-Div quantifies the applicability of the hypothesis $\widehat{h}_{\widehat{\mathcal{P}}}$ selected by the algorithm $\mathcal{A}$ from the training dataset $\widehat{\mathcal{P}}$ to the test dataset $\widehat{\mathcal{Q}}$. Specifically, I-Div employs the empirical estimator in Eq. (8) with hypothesis-oriented density ratio $\widehat{r}$ in Eq. (14), and adaptive likelihood ratio $\widehat{v}$ in Eq. (17) to estimate the distribution between training and test distributions by

$$\widehat{d}\left(\widehat{\mathcal{P}}, \widehat{\mathcal{Q}} \mid \widehat{h}_{\widehat{\mathcal{P}}}, \widehat{r}, \widehat{v}\right) = \widehat{\mathbb{E}}_{\widehat{\mathcal{P}}}\left[|\widehat{r}(\mathbf{x})\,\widehat{v}(\mathbf{x}, y) - 1|\,\mathfrak{L}(\widehat{h}_{\widehat{\mathcal{P}}}(\mathbf{x}), y)\right], \tag{18}$$

where $\widehat{r}$ is chosen from the hypothesis space $\mathcal{R}(\widehat{h}_{\widehat{\mathcal{P}}})$ based on $\widehat{h}_{\widehat{\mathcal{P}}}$, and $\widehat{v}$ is selected from the space $\mathcal{V}(\widehat{r})$ based on $\widehat{r}$. A smaller discrepancy indicates that the training and test samples are likely drawn from the same distribution relative to $\widehat{h}_{\widehat{\mathcal{P}}}$. Since $\widehat{h}_{\widehat{\mathcal{P}}}$ minimizes the empirical risk on $\widehat{\mathcal{P}}$, a reduced distribution discrepancy improves the transferability of the hypothesis from training to test samples. On the other hand, a greater discrepancy suggests a reduced likelihood of hypothesis applicability. The I-Div methodology is detailed in Algorithm 1.

---

**Algorithm 1** Importance divergence

---

1: **Input:**
2:     - Training samples $\widehat{\mathcal{P}} = \{(\mathbf{x}_i, y_i, c_i = 1)\}_{i=1}^{N} \sim \mathcal{P}$
3:     - Test samples $\widehat{\mathcal{Q}} = \{(\mathbf{x}_i, c_i = -1)\}_{i=1}^{N} \sim \mathcal{Q}$
4:     - Empirical minimizer $\widehat{h}_{\widehat{\mathcal{P}}}$, Hyperparameters $\lambda$ and $\gamma$
5: Merge datasets: $\widehat{\mathcal{U}} = \widehat{\mathcal{P}} \cup \widehat{\mathcal{Q}}$
6: Estimate the hypothesis-oriented density ratio on $\widehat{\mathcal{U}}$:

$$\widehat{r} \in \arg\min_{r \in \mathcal{R}(\widehat{h}_{\widehat{\mathcal{P}}})} \widehat{\mathbb{E}}_{(\mathbf{x},c)\sim\widehat{\mathcal{U}}}\left[c \log r(\mathbf{x}) + \lambda \left\|(r(\mathbf{x}))^c - 1\right\|^2\right]$$

7: Estimate the adaptive likelihood ratio on $\widehat{\mathcal{P}}$:

$$\widehat{v}(\mathbf{x}, y) = \left(N\gamma\widehat{r}(\mathbf{x}) - N\right) / \left(\gamma(\widehat{r}(\mathbf{x}))^2\right) \cdot \widehat{\mathbb{E}}_{\widetilde{\mathbf{x}}\in\widehat{\mathcal{P}}}\left[\left(\gamma(\widehat{r}(\widetilde{\mathbf{x}}))^2\right) / \left(\gamma\widehat{r}(\widetilde{\mathbf{x}}) - 1\right)\right]$$

8: Estimate the distribution discrepancy with importance sampling on $\widehat{\mathcal{P}}$:

$$\widehat{d}\left(\widehat{\mathcal{P}}, \widehat{\mathcal{Q}} \mid \widehat{h}_{\widehat{\mathcal{P}}}, \widehat{r}, \widehat{v}\right) = \widehat{\mathbb{E}}_{\widehat{\mathcal{P}}}\left[|\widehat{r}(\mathbf{x})\,\widehat{v}(\mathbf{x}, y) - 1|\,\mathfrak{L}(\widehat{h}_{\widehat{\mathcal{P}}}(\mathbf{x}), y)\right]$$

9: **Output:** empirical estimator $\widehat{d}\left(\widehat{\mathcal{P}}, \widehat{\mathcal{Q}} \mid \widehat{h}_{\widehat{\mathcal{P}}}, \widehat{r}, \widehat{v}\right)$

---

## 4 Experimental results

This section presents a comparative analysis of I-Div [2] against existing methods for evaluating the distribution discrepancy between training and test samples. The detailed experimental setups are presented in Appendix D.1.

---

[2]The source code is publicly available at: `https://github.com/Lawliet-zzl/I-div`.

Table 1: Distribution discrepancy of different classes in CIFAR10. The larger the values of AUROC and AUPR, the better the performance.

| DATASET | TARGET | MSP | | NNBD | | MMD-D | | R-DIV | | I-DIV | |
|---|---|---|---|---|---|---|---|---|---|---|---|
| | | AUROC | AUPR | AUROC | AUPR | AUROC | AUPR | AUROC | AUPR | AUROC | AUPR |
| CIFAR10 | AIRPLANE | 100.0 | 100.0 | 93.1 | 93.4 | 97.5 | 97.6 | 100.0 | 100.0 | 100.0 | 100.0 |
| | AUTOMOBILE | 100.0 | 100.0 | 96.5 | 96.2 | 93.6 | 94.5 | 100.0 | 100.0 | 100.0 | 100.0 |
| | BIRD | 100.0 | 100.0 | 90.4 | 90.0 | 97.2 | 97.6 | 100.0 | 100.0 | 100.0 | 100.0 |
| | CAT | 100.0 | 100.0 | 94.0 | 93.9 | 86.9 | 87.9 | 100.0 | 100.0 | 100.0 | 100.0 |
| | DEER | 100.0 | 100.0 | 90.9 | 90.8 | 91.7 | 92.1 | 100.0 | 100.0 | 100.0 | 100.0 |
| | DOG | 100.0 | 100.0 | 95.5 | 95.3 | 95.9 | 96.4 | 100.0 | 100.0 | 100.0 | 100.0 |
| | FROG | 100.0 | 100.0 | 91.7 | 91.6 | 96.0 | 96.5 | 100.0 | 100.0 | 100.0 | 100.0 |
| | HORSE | 100.0 | 100.0 | 91.9 | 91.8 | 82.8 | 83.4 | 100.0 | 100.0 | 100.0 | 100.0 |
| | SHIP | 100.0 | 100.0 | 95.6 | 95.3 | 98.7 | 98.9 | 100.0 | 100.0 | 100.0 | 100.0 |
| | TRUCK | 100.0 | 100.0 | 96.9 | 96.7 | 90.8 | 91.9 | 100.0 | 100.0 | 100.0 | 100.0 |

## 4.1 Experiments on different classes

Our initial experiments focus on a relatively straightforward task: assessing the applicability of a hypothesis obtained on the training dataset to the test dataset with distinctly different class labels. We utilize two datasets, CIFAR10 [31] and SVHN [32], each comprising ten semantically unique classes. For our experiments, we select samples from one class to serve as the test dataset, with the samples from the remaining nine classes forming the training dataset. This setup clearly illustrates that the knowledge learned in the training dataset cannot be transferred to the test dataset.

The results for CIFAR10 and SVHN are detailed in Table 1 and Table 6 (Appendix D.2), respectively. Our proposed I-Div algorithm consistently achieves perfect scores ($100\%$) in both AUROC and AUPR metrics across all classes of both datasets. This demonstrates its exceptional capability in distinguishing between training and test datasets, aligned with our initial hypothesis. The results unequivocally support the premise that the knowledge transfer from the training to the test datasets is ineffective, as evidenced by the flawless performance of I-Div. This starkly contrasts with the varying effectiveness of other algorithms, including NNBD and MMD-D. Notably, I-Div, MSP, and R-Div all yielded similarly impressive results. A key commonality among these algorithms is their reliance on a specific hypothesis to calculate distribution discrepancy, as opposed to NNBD and MMD-D, which use independent hypotheses. This highlights the significance of considering a particular hypothesis when evaluating distribution discrepancies. The rationale is that the hypothesis applicability depends on the specific design and its intended task.

## 4.2 Experiments on different datasets

We now turn to a more complex scenario where the training and test datasets may share semantic similarities in class labels, indicating an overlap in the class label spaces. In cases where semantics differ significantly, we expect the algorithm to clearly differentiate the two kinds of samples. Conversely, if their semantics are similar, the algorithm may find it challenging to

Table 2: Distribution discrepancy of domain adaptation data.

| DATASET | SOURCE | ACC | AUROC | | | | |
|---|---|---|---|---|---|---|---|
| | | | MSP | NNBD | MMD-D | R-DIV | I-DIV |
| PACS | P | 94.7 | 100.0 | 99.4 | 95.8 | 100.0 | 39.7 |
| | A | 77.4 | 100.0 | 98.3 | 96.9 | 100.0 | 42.1 |
| | C | 74.3 | 100.0 | 98.2 | 95.2 | 100.0 | 41.5 |
| | S | 78.9 | 100.0 | 99.6 | 94.6 | 100.0 | 49.5 |
| OFFICE-HOME | P | 76.1 | 100.0 | 97.2 | 94.6 | 100.0 | 44.8 |
| | A | 58.6 | 100.0 | 98.7 | 96.5 | 100.0 | 48.0 |
| | C | 48.5 | 100.0 | 98.8 | 95.8 | 100.0 | 51.4 |
| | R | 74.1 | 100.0 | 97.8 | 95.8 | 100.0 | 49.5 |

distinguish them. This outcome would suggest that the knowledge acquired from the training dataset is transferable to the test dataset, or it may indicate potential pathways to enhance the hypothesis generalization for the test distribution. We leverage CLIP [33] to align class labels between the training and test datasets, using the prompt template "A photo of a {label}." This helps adapt class labels across domains and captures semantic relationships between different datasets.

We conduct experiments using classic domain adaptation datasets: PACS [34] and Office-Home [35], each containing four domains. We designate one domain as the training dataset and merge the remaining three as the test dataset. The results, presented in Table 2, indicate that each hypothesis selected from a dataset performs significantly better than randomly selected hypotheses, demonstrating its applicability. Our I-Div algorithm aptly reflects this, in contrast to other algorithms that overly emphasize distribution discrepancies, thereby rigidly categorizing the difference between training and test datasets.

Table 3: Distribution discrepancy of different datasets on ResNet18. For CIFAR10.1 and STL10, smaller values of AUROC and AUPR indicate better performance. However, for other test datasets, larger values are better.

| TARGET | ACC (CLIP) | MSP | | NNBD | | MMD-D | | R-DIV | | I-DIV | |
|---|---|---|---|---|---|---|---|---|---|---|---|
| | | AUROC | AUPR | AUROC | AUPR | AUROC | AUPR | AUROC | AUPR | AUROC | AUPR |
| RGI | 0.0 | **100.0** | **100.0** | **100.0** | **100.0** | 99.2 | 99.3 | **100.0** | **100.0** | **100.0** | **100.0** |
| SVHN | 17.0 | **100.0** | **100.0** | **100.0** | **100.0** | 94.8 | 95.7 | **100.0** | **100.0** | **100.0** | **100.0** |
| DTD | 1.9 | **100.0** | **100.0** | **100.0** | **100.0** | 100.0 | 100.0 | **100.0** | **100.0** | **100.0** | **100.0** |
| FLOWERS102 | 1.6 | **100.0** | **100.0** | **100.0** | **100.0** | 97.0 | 98.8 | **100.0** | **100.0** | **100.0** | **100.0** |
| OXFORDIIITPET | 2.3 | **100.0** | **100.0** | **100.0** | **100.0** | 98.7 | 100.0 | **100.0** | **100.0** | **100.0** | **100.0** |
| SEMEION | 8.7 | **100.0** | **100.0** | **100.0** | **100.0** | 98.5 | 99.0 | **100.0** | **100.0** | **100.0** | **100.0** |
| CALTECH256 | 2.4 | **100.0** | **100.0** | **100.0** | **100.0** | 92.3 | 92.4 | **100.0** | **100.0** | 99.9 | 99.9 |
| CIFAR100 | 2.2 | **100.0** | **100.0** | 99.9 | 99.9 | 90.2 | 91.5 | **100.0** | **100.0** | 94.6 | 94.7 |
| CIFAR10.1 | **73.4** | 100.0 | 100.0 | 92.1 | 93.7 | 92.9 | 93.1 | 100.0 | 100.0 | **43.4** | **45.2** |
| STL10 | **63.0** | 100.0 | 100.0 | 94.0 | 93.3 | 90.4 | 91.1 | 100.0 | 100.0 | **37.2** | **41.9** |

We use CIFAR10 [31] as the training dataset and evaluate on diverse test datasets including Randomly Generated Images (RGI), SVHN [32], DTD [36], Flowers102 [37], OxfordIIITPet [38], SEMEION [39], Caltech256 [40], CIFAR100 [31], CIFAR10.1 [41], and STL10. Since CIFAR10.1 and STL10 share similar category spaces with CIFAR10, the model shows minimal differentiation for these datasets, with ACC values of 73.4% and 42.1%, respectively. This suggests partial knowledge transferability. However, I-Div demonstrates lower AUROC values of 43.7% and 37.2%, indicating reduced discrimination. Other algorithms like MSP and R-Div show higher AUROC values, near 100% across all datasets, but these results suggest an overemphasis on distribution discrepancy rather than semantic similarity. I-Div, in contrast, better captures semantic relationships between datasets, providing a more nuanced view of class label semantics.

Table 4: Distribution discrepancy between ImageNet and other test datasets.

| TRAINING | NETWORK | TEST | ACC (CLIP) | MSP | NNBD | MMD-D | H-DIV | R-DIV | I-DIV |
|---|---|---|---|---|---|---|---|---|---|
| IMAGENET | RESNET50 | OIDv4 | 43.9 | 100.0 | 91.7 | 94.6 | 100.0 | 94.6 | **69.3** |
| | | CALTECH256 | 36.6 | 100.0 | 91.4 | 95.6 | 100.0 | 100.0 | **72.4** |
| | | FLOWERS102 | 5.1 | 100.0 | 98.6 | 100.0 | 100.0 | 100.0 | **100.0** |
| | | DTD | 11.9 | 100.0 | 98.7 | 100.0 | 100.0 | 100.0 | **100.0** |
| | VIT-B/16 | OIDv4 | 50.6 | 100.0 | 88.6 | 92.6 | 100.0 | 92.6 | **62.6** |
| | | CALTECH256 | 40.4 | 100.0 | 94.8 | 100.0 | 100.0 | 100.0 | **71.9** |
| | | FLOWERS102 | 5.1 | 100.0 | 98.1 | 100.0 | 100.0 | 100.0 | **100.0** |
| | | DTD | 13.9 | 100.0 | 99.7 | 100.0 | 100.0 | 100.0 | **100.0** |

We use ImageNet [42] as the training dataset and evaluate on diverse test datasets using ResNet50 [43] and ViT-B/16 [44]. The test datasets include the Open Images Dataset v4 (OIDv4) [45], Caltech256 [40], Flowers102 [37], and DTD [36]. The experimental results presented in Table 4 show that the AUROC values of I-Div effectively capture the semantic similarity between ImageNet and the test datasets, yielding results that closely align with human intuition. For example, I-Div demonstrates lower AUROC values for OID and Caltech256, reflecting their semantic overlap with ImageNet, as these datasets share common object categories and scene types. In contrast, datasets such as Flowers102 and DTD, which focus on more specialized object categories and textures, show higher AUROC values with I-Div, indicating greater divergence from ImageNet. On the other hand, algorithms like MSP, NNBD, and MMD-D show consistently high AUROC values across most datasets, implying they emphasize distribution discrepancies over semantic relationships. This limits their effectiveness in distinguishing nuanced semantic differences compared to I-Div, which provides a more human-aligned understanding of dataset relationships.

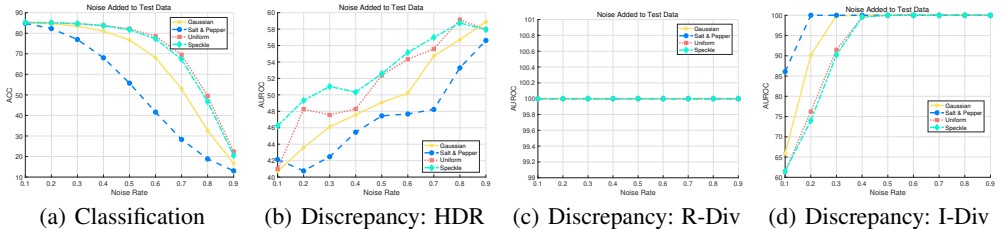

| (a) Classification | (b) Discrepancy: HDR | (c) Discrepancy: R-Div | (d) Discrepancy: I-Div |

Figure 1: Distribution discrepancy between original data and its corrupted variants with different noise rate. (a) shows the classification performance of the standard network for the test datasets containing corrupted samples. (b)(c)(d) present the distribution discrepancy in terms of AUROC.

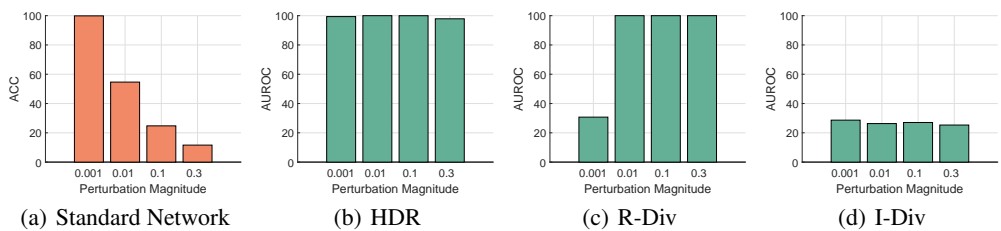

| (a) Standard Network | (b) HDR | (c) R-Div | (d) I-Div |

Figure 2: Distribution discrepancy between original data and adversarial data.

## 4.3 Experiments on corrupted data

This section discusses experimental results on corrupted datasets. We progressively introduce noise into a dataset that serves as the training one, treating the resultant corrupted samples as the test dataset. Intuitively, as the noise level increases, the hypothesis performs worse, which indicates the hypothesis becomes less applicable to the corrupted data. To conduct this experiment, CIFAR10 serves as the training dataset, with incremental addition of noises to the original dataset to create the test dataset. The types of noises [46] used include Gaussian, Salt & Pepper, Uniform, and Speckle, with the noise rate increasing from 0.1 to 0.9 with a 0.1 interval. The methods for comparison include Hypothesis-oriented Density Ratio (HDR) in I-Div and R-Div [47]. Fig. 1 presents our experimental findings, showing key performance metrics as influenced by varying noise rates. Notably, the classification performance declines with increasing noise, impacting the hypothesis predictive accuracy. Interestingly, our proposed I-Div algorithm demonstrates robustness against these challenges, with its discrimination power inversely related to the classification accuracy of the standard network in noisy conditions. A brief comparative analysis hints at the superior performance of I-Div over HDR, especially in relation to hypothesis applicability in noisy test datasets. Fig. 4 in Appendix D.3 shows the results when noise is added to the training data instead of the test data. The results are consistent with the above, as the classification accuracy decreases with increasing noise, and I-Div becomes more effective in distinguishing between clean training data and noisy test data. For a comprehensive discussion and full experimental results, please see Appendix D.3.

## 4.4 Experiments on adversarial data

In this experiment, we delve into a specific scenario involving adversarial samples [48]. We designate one dataset as training and its corresponding adversarial samples as the test dataset. It is a well-known phenomenon that a minimal adversarial perturbation, though visually imperceptible, can drastically alter the classification performance of a network. This suggests potential issues with the direct applicability of the hypothesis selected for a training dataset to a test dataset. However, based on human perception, which fails to distinguish original and adversarial samples visually, we would expect a negligible distribution discrepancy between the distributions of the original and adversarial samples. This outcome could guide us in enhancing network robustness against adversarial attacks and in generalizing the hypothesis to adversarial contexts. For this purpose, we use the CIFAR10 dataset to train a standard network, with adversarial perturbation magnitudes selected from the set $\{0.001, 0.01, 0.1, 0.3\}$. The results in Fig. 2 indicate a marked decrease in standard network

Table 5: Effect of different network architectures.

| TARGET | RESNET18 | | | VGG19 | | | MOBILENET | | | EFFICIENTNET | | |
|---|---|---|---|---|---|---|---|---|---|---|---|---|
| | ACC | AUROC | AUPR | ACC | AUROC | AUPR | ACC | AUROC | AUPR | ACC | AUROC | AUPR |
| RGI | 0.0 | 100.0 | 100.0 | 0.0 | 73.0 | 70.0 | 0.0 | 100.0 | 100.0 | 0.0 | 99.6 | 98.2 |
| SVHN | 17.0 | 100.0 | 100.0 | 15.6 | 74.5 | 69.8 | 20.0 | 100.0 | 100.0 | 19.8 | 98.1 | 93.7 |
| DTD | 1.9 | 100.0 | 100.0 | 2.3 | 76.3 | 71.6 | 2.5 | 100.0 | 100.0 | 1.8 | 97.0 | 90.8 |
| FLOWERS102 | 1.6 | 100.0 | 100.0 | 2.0 | 78.3 | 73.5 | 2.4 | 100.0 | 100.0 | 2.7 | 96.7 | 89.8 |
| OXFORDIIITPET | 2.3 | 100.0 | 100.0 | 0.9 | 69.8 | 67.3 | 1.6 | 100.0 | 100.0 | 1.7 | 97.4 | 91.8 |
| SEMEION | 8.7 | 100.0 | 100.0 | 7.8 | 78.4 | 73.3 | 9.6 | 100.0 | 100.0 | 10.4 | 98.1 | 93.5 |
| CALTECH256 | 2.4 | 99.9 | 99.9 | 2.5 | 72.4 | 67.0 | 2.3 | 99.2 | 99.2 | 2.0 | 96.8 | 94.3 |
| CIFAR100 | 2.2 | 94.6 | 94.7 | 2.6 | 66.0 | 60.3 | 2.9 | 83.3 | 83.0 | 1.9 | 89.5 | 79.7 |
| CIFAR101 | 73.4 | 43.4 | 45.2 | 82.5 | 34.8 | 40.7 | 70.0 | 45.8 | 47.1 | 74.2 | 44.4 | 49.8 |
| STL10 | 63.0 | 37.2 | 41.9 | 72.0 | 50.1 | 48.5 | 60.6 | 43.8 | 31.1 | 63.9 | 39.5 | 47.5 |

accuracy against adversarial perturbations, while the consistent AUROC of I-Div suggests its limited differentiation capability. Detailed results and analyses are provided in Appendix D.4.

### 4.5 Experiments with different sample sizes and network architectures

We examine the impact of different sample sizes on the performance of the I-Div algorithm, focusing on its ability to generalize hypotheses from training to test datasets. Fig. 3 show that I-Div tends to maintain low AU-ROC values for semantically similar datasets like CIFAR10.1 and STL10, indicating effective hypothesis applicability. Conversely, for datasets with significant semantic differences, the performance of I-Div improves with larger sample sizes, highlighting its

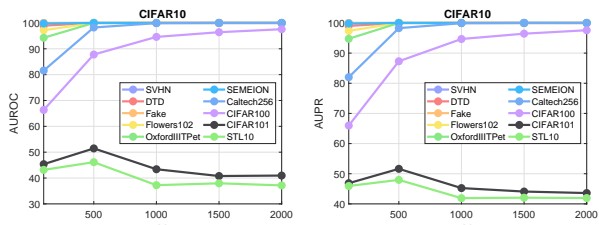

Figure 3: Effect of different sample sizes.

capacity to recognize non-transferable knowledge. Additionally, we investigate the effect of varying network architectures as shown in Table 5. Detailed results are provided in Appendix D.5.

## 5 Limitations

I-Div relies on density and likelihood ratios to achieve the sampling transfer in unlabeled test data, allowing for the estimation of distribution discrepancies between training and test datasets with labeled training samples. Although the density ratio can be accurately estimated using inputs from both training and test samples, the likelihood ratio cannot be estimated precisely due to the unavailability of class labels of test samples. The strategy used by I-Div targets the estimation of distribution discrepancies between training and test distributions. It optimizes a likelihood ratio that adapts to this density ratio to ensure a rapid convergence of the distribution discrepancy, by minimizing the upper bound of the generalization error based on the density ratio. Our future research includes refining the estimation methods for likelihood ratios and exploring distribution discrepancy estimation methods that can bypass the likelihood ratio.

## 6 Conclusion

In the realm of complex data and machine learning tasks, a crucial question arises regarding the applicability of a hypothesis derived from a training dataset to a test dataset. This uncertainty, especially challenging when test samples lack class labels, significantly determining the hypothesis generalization. To address this, we introduce the I-Div measure for estimating the distribution discrepancy between training and test distributions. I-Div involves the hypothesis-oriented density ratio and adaptive likelihood ratio in expected risk difference to shift the sampling problem from test to training distributions. Experimentally, we validate that I-Div can effectively assess the hypothesis capability of handling test samples, yielding results consistent with prior human knowledge.

## Acknowledgments and Disclosure of Funding

This work was supported in part by the Australian Research Council Linkage Grant LP230201022, the Australian Research Council Discovery Grant DP240102050, and the Australian Research Council Linkage Infrastructure, Equipment and Facilities Grant LE240100131.

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

# A  Related work

In this section, we briefly review the methods for distribution discrepancy estimation, density ratio estimation, and performance prediction.

## A.1  Distribution discrepancy estimation

Mean Embedding (ME) [49] and Smooth Characteristic Functions (SCF) [50] utilize differences in Gaussian mean embeddings at optimized points and frequencies, respectively, to quantify distribution discrepancy. Building on ME, MMD-O [51] calculates Maximum Mean Discrepancy (MMD) using a Gaussian kernel [52], while MMD-D [53] enhances the performance of MMD-O by substituting the Gaussian Kernel with a learnable deep kernel. In contrast, Classifier Two-Sample Tests, including C2ST-S [54] and C2ST-L [55], classify samples from one dataset as positive and from another as negative, leveraging the classification accuracy of a binary classifier to differentiate between them. H-Divergence (H-Div) [20] identifies optimal hypotheses for both the mixture distribution and individual distributions within a specific model, positing that if two distributions are identical, the expected risk on the training samples from the mixture distribution exceeds that from each individual distribution. R-Divergence (R-Div) [47] tackles the overfitting problem of H-Div by suggesting that two distributions are likely identical if their optimal hypotheses yield the same expected risk for each. However, to estimate the discrepancy between training and test distributions, the aforementioned methods assume symmetry between these datasets, meaning either both sets are labeled or both are unlabeled.

## A.2  Density ratio estimation

To estimate the density ratio, Kernel Mean Matching (KMM) [56] offers direct estimates of importance at training inputs by efficiently aligning two distributions, leveraging a unique characteristic of universal reproducing kernel Hilbert spaces. The Kullback-Leibler Importance Estimation Procedure (KLIEP) [19] seeks an importance estimate to minimize the Kullback-Leibler divergence from the true test input density to its estimate. Least-Squares Importance Fitting (LSIF) [57] addresses the direct importance estimation issue as a least-squares function fitting problem, transforming the optimization challenge into a convex quadratic program, which is solvable efficiently with standard quadratic programming solvers. Non-Negative Bregman Divergence (NNBD) [58] employs deep neural networks alongside empirical Bregman divergence minimization, addressing the train-loss hacking issue by adjusting the empirical Bregman divergence estimator. Although these methods can estimate the distribution discrepancy between test and training inputs by calculating the density ratio even when class labels for test samples are not available, this implies disregarding the class labels of the training data. Consequently, such density ratios are not suitable for evaluating the applicability of a hypothesis.

## A.3  Performance prediction

Maximum over Softmax Probability (MSP) [59] gauges the class label confidence on each sample by a given model to distinguish ID and OOD samples, which can be extend by computing the average class label confidence across both training and test datasets and consider the confidence disparity as indicative of distribution discrepancy. Average Thresholded Confidence (ATC) [60] identifies a threshold such that the proportion of training samples exceeding this confidence threshold aligns with their accuracy, and then it predicts accuracy as the proportion of unlabeled test samples surpassing this threshold. Agreement-on-the-Line (AL) [61] capitalizes on the observation that if the accuracy of models on training samples linearly correlates with their accuracy on test samples, then a similar linear relationship exists between the training and test agreement of model predictions. Projection Norm (PN) [62] employs model predictions to pseudo-label test samples and trains a new model on these pseudo-labels. The variation in parameters between this new model and the original model is utilized to estimate the test error. The aforementioned algorithms estimate the effectiveness of a hypothesis selected from the training samples on test samples based on the confidence output of the hypothesis to judge its applicability. However, since the hypothesis is chosen based on a limited set of training samples, it exhibits significant uncertainty towards unseen test samples, especially when these test samples are OOD, leading to misleading results.

# B Objective function for estimating density ratio

According to the KL divergence between $\mathcal{P}(\mathbf{x})$ and $\widetilde{\mathcal{P}}(\mathbf{x})$, we have

$$\mathrm{KL}\left(\mathcal{P}(\mathbf{x}) \parallel \widetilde{\mathcal{P}}(\mathbf{x})\right) = \int_{x \in \mathcal{X}} \mathcal{P}(\mathbf{x}) \log \frac{\mathcal{P}(\mathbf{x}) r(\mathbf{x})}{\mathcal{Q}(\mathbf{x})} \, d\mathbf{x} = \mathrm{KL}\left(\mathcal{P}(\mathbf{x}) \parallel \mathcal{Q}(\mathbf{x})\right) + \mathbb{E}_{\mathcal{P}}\left[\log r(\mathbf{x})\right]. \tag{19}$$

Similarly, for the KL divergence between $\mathcal{Q}(\mathbf{x})$ and $\widetilde{\mathcal{Q}}(\mathbf{x})$, we have

$$\mathrm{KL}\left(\mathcal{Q}(\mathbf{x}) \parallel \widetilde{\mathcal{Q}}(\mathbf{x})\right) = \int_{\mathbf{x} \in \mathcal{X}} \mathcal{Q}(\mathbf{x}) \log \frac{\mathcal{Q}(\mathbf{x})}{r(\mathbf{x}) \mathcal{P}(\mathbf{x})} \, d\mathbf{x} = \mathrm{KL}\left(\mathcal{Q}(\mathbf{x}) \parallel \mathcal{P}(\mathbf{x})\right) - \mathbb{E}_{\mathcal{Q}}\left[\log r(\mathbf{x})\right]. \tag{20}$$

Note that the terms $\mathrm{KL}\left(\mathcal{P}(\mathbf{x}) \parallel \widetilde{\mathcal{P}}(\mathbf{x})\right)$ and $\mathrm{KL}\left(\mathcal{Q}(\mathbf{x}) \parallel \widetilde{\mathcal{Q}}(\mathbf{x})\right)$ are constants independent of the density ratio $r$. To normalize $\widetilde{\mathcal{P}}(\mathbf{x})$ and $\widetilde{\mathcal{Q}}(\mathbf{x})$ estimated by $r$, we have $\int_{\mathbf{x} \in \mathcal{X}} \widetilde{\mathcal{P}}(\mathbf{x}) \, d\mathbf{x} = 1$ and $\int_{\mathbf{x} \in \mathcal{X}} \widetilde{\mathcal{Q}}(\mathbf{x}) \, d\mathbf{x} = 1$. According to Eq. (10), we have

$$\frac{1}{2} \int_{\mathbf{x} \in \mathcal{X}} \frac{\mathcal{Q}(\mathbf{x})}{r(\mathbf{x})} \, d\mathbf{x} + \frac{1}{2} \int_{\mathbf{x} \in \mathcal{X}} r(\mathbf{x}) \mathcal{P}(\mathbf{x}) \, d\mathbf{x} = 1. \tag{21}$$

By considering Eq. (19), Eq. (20) and Eq. (21), we can obtain an optimization problem with constraints for learning a density ratio $r$, i.e.,

$$\begin{aligned} \min_{r \in \mathcal{R}(\widehat{h}_{\widehat{\mathcal{P}}})} \quad & \mathbb{E}_{\mathcal{P}}\left[\log r(\mathbf{x})\right] + \mathbb{E}_{\mathcal{Q}}\left[-\log r(\mathbf{x})\right] \\ \text{s.t.} \quad & \frac{1}{2} \int_{\mathbf{x} \in \mathcal{X}} \frac{\mathcal{Q}(\mathbf{x})}{r(\mathbf{x})} \, d\mathbf{x} + \frac{1}{2} \int_{\mathbf{x} \in \mathcal{X}} r(\mathbf{x}) \mathcal{P}(\mathbf{x}) \, d\mathbf{x} = 1. \end{aligned} \tag{22}$$

However, solving such a problem with constraints is difficult. Inspired by the method of Lagrange multipliers [30], we can introduce a hyperparameter $\lambda \geq 0$, which allows us to relax this constraint and balance the loss function of $r$ with the constraint conditions, i.e.,

$$\begin{aligned} \min_{r \in \mathcal{R}(\widehat{h}_{\widehat{\mathcal{P}}})} \quad & \mathbb{E}_{\mathcal{P}}\left[\log r(\mathbf{x})\right] + \mathbb{E}_{\mathcal{Q}}\left[-\log r(\mathbf{x})\right] + \lambda \mathcal{J}. \\ \text{s.t.} \quad & \mathcal{J} = \frac{1}{2} \left\| \frac{1}{2} \int_{\mathbf{x} \in \mathcal{X}} \frac{\mathcal{Q}(\mathbf{x})}{r(\mathbf{x})} \, d\mathbf{x} + \frac{1}{2} \int_{\mathbf{x} \in \mathcal{X}} r(\mathbf{x}) \mathcal{P}(\mathbf{x}) \, d\mathbf{x} - 1 \right\|^2. \end{aligned} \tag{23}$$

Accordingly to the Cauchy-Schwarz inequality [63], we have

$$\begin{aligned} \sqrt{\mathcal{J}} &\leq \left\| \int_{\mathbf{x} \in \mathcal{X}} \frac{\mathcal{Q}(\mathbf{x})}{r(\mathbf{x})} \, d\mathbf{x} - 1 \right\| + \left\| \int_{\mathbf{x} \in \mathcal{X}} r(\mathbf{x}) \mathcal{P}(\mathbf{x}) \, d\mathbf{x} - 1 \right\| \\ &\leq \int_{\mathbf{x} \in \mathcal{X}} \mathcal{Q}(\mathbf{x}) \left\| \frac{1}{r(\mathbf{x})} - 1 \right\| \, d\mathbf{x} + \int_{\mathbf{x} \in \mathcal{X}} \mathcal{P}(\mathbf{x}) \left\| r(\mathbf{x}) - 1 \right\| \, d\mathbf{x} \\ &= \mathbb{E}_{\mathcal{Q}} \left\| \frac{1}{r(\mathbf{x})} - 1 \right\| + \mathbb{E}_{\mathcal{P}} \left\| r(\mathbf{x}) - 1 \right\|. \end{aligned} \tag{24}$$

Applying Eq. (24) and the Jensen's inequality, we have

$$\begin{aligned} \mathcal{J} &\leq \left( \mathbb{E}_{\mathcal{Q}} \left\| \frac{1}{r(\mathbf{x})} - 1 \right\| \right)^2 + \left( \mathbb{E}_{\mathcal{P}} \left\| r(\mathbf{x}) - 1 \right\| \right)^2 \\ &\leq \mathbb{E}_{\mathcal{Q}} \left[ \left\| \frac{1}{r(\mathbf{x})} - 1 \right\|^2 \right] + \mathbb{E}_{\mathcal{P}} \left[ \left\| r(\mathbf{x}) - 1 \right\|^2 \right]. \end{aligned} \tag{25}$$

We obtain the objective function by combining Eq. (23) and Eq. (25).

## C  Proofs

### C.1  Proof of Theorem 3.1

We rewrite $f(r)$ by applying the density ratio and obtain

$$
\begin{aligned}
f(r) =& \frac{1}{2}\mathbb{E}_{\mathcal{P}}\left[\log r(\mathbf{x}) + \lambda\left\|r(\mathbf{x}) - 1\right\|^2\right] + \frac{1}{2}\mathbb{E}_{\mathcal{Q}}\left[\lambda\left\|\frac{1}{r(\mathbf{x})} - 1\right\|^2 - \log r(\mathbf{x})\right] \\
=& \frac{1}{2}\mathbb{E}_{\mathcal{P}}\left[\log r(\mathbf{x}) + \lambda\left\|r(\mathbf{x}) - 1\right\|^2\right] + \frac{1}{2}\mathbb{E}_{\mathcal{P}}\left[\lambda r(\mathbf{x})\left\|\frac{1}{r(\mathbf{x})} - 1\right\|^2 - r(\mathbf{x})\log r(\mathbf{x})\right] \quad (26) \\
=& \frac{1}{2}\mathbb{E}_{\mathcal{P}}\left[\log r(\mathbf{x}) + \lambda\left\|r(\mathbf{x}) - 1\right\|^2 + \lambda r(\mathbf{x})\left\|\frac{1}{r(\mathbf{x})} - 1\right\|^2 - r(\mathbf{x})\log r(\mathbf{x})\right].
\end{aligned}
$$

According to Eq. (12), we know that $\lambda$ is used to ensure the constraints are satisfied as much as possible. Intuitively, its value should be relatively large. More specifically, we provide the following lemma to determine the lower bound of $\lambda$ to ensure that $f(r)$ is strongly convex.

**Lemma C.1.**  $f(r)$ is $\mu$-strongly convex if $\lambda \geq 1$.

Thus, we have

$$
\begin{aligned}
\frac{\mu}{2}\mathbb{E}_{\mathcal{P}}\left|\widehat{r}(\mathbf{x}) - \bar{r}(\mathbf{x})\right|^2 \leq& f(\widehat{r}) - f(\bar{r}) = f(\widehat{r}) - \widehat{f}(\bar{r}) + \widehat{f}(\bar{r}) - f(\bar{r}) \\
\leq& f(\widehat{r}) - \widehat{f}(\widehat{r}) + \widehat{f}(\bar{r}) - f(\bar{r}) \leq 2\sup_{r \in \mathcal{R}(h)}\left|f(r) - \widehat{f}(r)\right|.
\end{aligned}
\qquad (27)
$$

The first inequality arises from the $\mu$-strong convexity of $u(r(\mathbf{x}))$. The second inequality is because $\widehat{r}$ is the minimizer of $\widehat{f}(r)$. According to $r(\mathbf{x}) \in (b_r, B_r)$ and $f(r)$ is $L_f$-lipschitz continuous on $(b_r, B_r)$, we know that the absolute value of the difference caused by altering one data point in $|f(r) - \widehat{f}(r)|$ is bounded by $2L_f B_r$.

To proceed, we introduce the following two lemmas.

**Lemma C.2** ([21], Theorem 26.5).  *Assume that $\widehat{\mathcal{P}}$ contains $N$ i.i.d. samples from $\mathcal{P}$ and $|l(h, z)| \leq c$ for all $z$ and $h \in \mathcal{H}$. With probability of at least $1 - \delta$, for all $h \in \mathcal{H}$,*

$$
\left|\epsilon_{\mathcal{P}}(h) - \widehat{\epsilon}_{\widehat{\mathcal{P}}}(h)\right| \leq 2\mathfrak{R}(l \circ \mathcal{H} \circ \widehat{\mathcal{P}}) + 4c\sqrt{\frac{\ln(4/\delta)}{N}},
$$

*where $\mathfrak{R}(l \circ \mathcal{H} \circ \widehat{\mathcal{P}})$ is the Rademacher complexity of $l \circ \mathcal{H}$ with respect to $\widehat{\mathcal{P}}$.*

**Lemma C.3** ([29], Talagrand's Contraction Lemma).  *For any $L$-Lipschitz loss function $l(\cdot, \cdot)$ and hypothesis space $\mathcal{H}$, we obtain*

$$
\mathfrak{R}(l \circ \mathcal{H} \circ \widehat{\mathcal{P}}) \leq L\mathfrak{R}(\mathcal{H} \circ \widehat{\mathcal{P}}),
$$

*where $\mathfrak{R}(\mathcal{H} \circ \widehat{\mathcal{P}}) = \frac{1}{N}\mathbb{E}_{\boldsymbol{\sigma} \sim \{\pm 1\}^N}\left[\sup_{h \in \mathcal{H}}\sum_{x \in \widehat{\mathcal{P}}}\sigma_i h(x_i)\right]$ with random choice of $\boldsymbol{\sigma}$.*

Accordingly, with probability of at least $1 - \delta$, for any $r \in \mathcal{R}$, we have

$$
f(r) - \widehat{f}(r) \leq 2L_f\mathfrak{R}(\mathcal{R}(h) \circ \widehat{\mathcal{U}}) + 8L_f B_r\sqrt{\frac{\ln 4/\delta}{M}}.
\qquad (28)
$$

Recall that $\psi$ is a frozen network backbone and $\omega$ is a learnable adapter containing two fully connected layers and an activation layer in the middle. The activation function used in the middle and the last layers are GELU and Softplus, respectively. According to the Rademacher bound [64], we have

$$
\mathfrak{R}(\mathcal{R}(h) \circ \widehat{\mathcal{U}}) \leq 4\beta_1\beta_2\tau_1\tau_2\sqrt{\frac{O_\psi + 1}{2M}} + \frac{\beta_2\tau_2}{M}.
\qquad (29)
$$

We complete the proof by combining Jensen's inequality, Eq. (27), Eq. (28) and Eq. (29).

## C.2   Proof of Theorem 3.2

Applying the triangle inequality, for and any $v \in \mathcal{V}$, we have

$$
\left| d\left(\mathcal{P}, \mathcal{Q} \mid \widehat{h}_{\widehat{\mathcal{P}}}, \overline{r}, v\right) - \widehat{d}\left(\widehat{\mathcal{P}}, \widehat{\mathcal{Q}} \mid \widehat{h}_{\widehat{\mathcal{P}}}, \widehat{r}, v\right) \right|
$$
$$
\leq \underbrace{\left| \mathbb{E}_{\mathcal{P}}\left[ |\overline{r}(\mathbf{x})v(\mathbf{x},y) - 1| \, \mathfrak{L}\left(\widehat{h}_{\widehat{\mathcal{P}}}(\mathbf{x}),y\right)\right] - \mathbb{E}_{\mathcal{P}}\left[ |\widehat{r}(\mathbf{x})v(\mathbf{x},y) - 1| \, \mathfrak{L}\left(\widehat{h}_{\widehat{\mathcal{P}}}(\mathbf{x}),y\right)\right] \right|}_{\mathcal{B}_1}
$$
$$
+ \underbrace{\left| \mathbb{E}_{\mathcal{P}}\left[ |\widehat{r}(\mathbf{x})v(\mathbf{x},y) - 1| \, \mathfrak{L}\left(\widehat{h}_{\widehat{\mathcal{P}}}(\mathbf{x}),y\right)\right] - \widehat{\mathbb{E}}_{\widehat{\mathcal{P}}}\left[ |\widehat{r}(\mathbf{x})v(\mathbf{x},y) - 1| \, \mathfrak{L}\left(\widehat{h}_{\widehat{\mathcal{P}}}(\mathbf{x}),y\right)\right] \right|}_{\mathcal{B}_2}.
$$
$$(30)$$

To bound $\mathcal{B}_1$, with probability of at least $1 - \delta$, we have

$$
\mathcal{B}_1 = \mathbb{E}_{\mathcal{P}}\left[ \left( |\overline{r}(\mathbf{x})v(\mathbf{x},y) - 1| - |\widehat{r}(\mathbf{x})\, v(\mathbf{x},y) - 1| \right) \mathfrak{L}\left(\widehat{h}_{\widehat{\mathcal{P}}}(\mathbf{x}),y\right)\right]
$$
$$
\leq \mathbb{E}_{\mathcal{P}}\left[ v(\mathbf{x},y)\, |\overline{r}(\mathbf{x}) - \widehat{r}(\mathbf{x})|\, \mathfrak{L}\left(\widehat{h}_{\widehat{\mathcal{P}}}(\mathbf{x}),y\right)\right]
$$
$$
\leq B_{\mathfrak{L}} \mathbb{E}_{\mathcal{P}}\left[ v(\mathbf{x},y)\right] \mathbb{E}_{\mathcal{P}}\left[ |\overline{r}(\mathbf{x}) - \widehat{r}(\mathbf{x})|\right]
$$
$$
\leq 2 B_{\mathfrak{L}} \mathbb{E}_{\mathcal{P}}\left[ v(\mathbf{x},y)\right] \sqrt{\frac{\mathfrak{B}(\delta, N)}{\mu}},
$$
$$(31)$$

where the first inequality follows from the triangle inequality, the second is a consequence of Hölder's inequality, and the final inequality results from Theorem 3.1. Further, we know

$$
0 \leq |\widehat{r}(\mathbf{x})v(\mathbf{x},y) - 1| \, \mathfrak{L}\left(\widehat{h}_{\widehat{\mathcal{P}}}(\mathbf{x}),y\right) \leq |\widehat{r}(\mathbf{x})v(\mathbf{x},y) - 1| B_{\mathfrak{L}}.
$$
$$(32)$$

By applying Hoeffding's inequality [65], with probability of at least $1 - \delta$, we have

$$
\mathcal{B}_2 \leq B_{\mathfrak{L}} \sqrt{\frac{\ln(2/\delta) \sum_{(\mathbf{x},y) \sim \widehat{\mathcal{P}}} |\widehat{r}(\mathbf{x})v(\mathbf{x},y) - 1|^2}{N}}.
$$
$$(33)$$

We complete the proof by combining Eq. (30), Eq. (31), and Eq. (33).

## C.3   Proof of Lemma C.1

The second derivative of the function $f(r)$ with respect to $r$ is

$$
\nabla^2 f(r) = \mathbb{E}_{\mathcal{P}}\left[ \frac{2\lambda (r(\mathbf{x}))^3 - (r(\mathbf{x}))^2 - (r(\mathbf{x})) + 2}{2(r(\mathbf{x}))^3}\right] \geq \lambda - \mathbb{E}_{\mathcal{P}}\left[ \frac{(r(\mathbf{x}))^2 - (r(\mathbf{x})) + 2}{2(r(\mathbf{x}))^3}\right].
$$
$$(34)$$

We know that
$$
\frac{(r(\mathbf{x}))^2 - (r(\mathbf{x})) + 2}{2(r(\mathbf{x}))^3} \leq \frac{-3 + \sqrt{7} + (\sqrt{7} - 1)^2}{2(\sqrt{7} - 1)^3} \leq 0.27.
$$
$$(35)$$

To ensure $\mathcal{K}(r(\mathbf{x}))$ is strongly convex [66], we can simply assume $\lambda \geq 1$.

## D   Experiments

Our experiments are designed to align with established knowledge and intuition. We hypothesize that I-Div will indicate a minimal distribution discrepancy when the training and test distributions share semantically similar class labels, meaning that samples from both distributions can be treated as ID. This suggests the applicability of the hypothesis selected by an algorithm from the training samples to the test samples. Conversely, if class labels significantly differ in semantics, with samples from the training and test distributions being categorized as ID and OOD, respectively, we expect I-Div to reveal a more pronounced distribution discrepancy. This implies that the knowledge learned from the training samples may not be transferable to the test samples.

### D.1 Experimental setup

In our study, we explore whether a hypothesis selected from a training dataset retains its capacity when applied to a test dataset. Our quantitative analysis utilizes the I-Div metric to measure the distribution discrepancy between training and test distributions, pertinent to the hypothesis applicability. Unless otherwise specified, we set $\lambda = 1$ and $\gamma = 1$. We hypothesize that data with closer semantic relationships will exhibit smaller distribution discrepancies, as opposed to those with distinct semantic differences.

For the given training dataset $\widehat{\mathcal{P}}$ and test dataset $\widehat{\mathcal{Q}}$, we consider using subsets of these datasets to estimate the distribution discrepancy instead of the entire datasets. This is because using too many samples can trivialize the estimation task and, realistically, obtaining the entire test dataset at once is impractical. Accordingly, we randomly draw samples from $\widehat{\mathcal{P}}$ and $\widehat{\mathcal{Q}}$, creating a subset of three smaller datasets $(\widehat{\mathcal{P}}', \widehat{\mathcal{P}}'', \widehat{\mathcal{Q}}')$. Adhering to the two-sample test framework [53, 67], each subset consists of $M \ll N$ samples. We then form positive pairings $(\widehat{\mathcal{P}}', \widehat{\mathcal{P}}'')$ and negative pairings $(\widehat{\mathcal{P}}', \widehat{\mathcal{Q}}')$. Unless otherwise noted, our experiments use a standard sample size of $M = 1000$. We generate $100,000$ tuples to achieve significant distribution discrepancy in the positive pairs and minimal discrepancy in the negative pairs.

The effectiveness is evaluated using the Area Under the Receiver Operating Characteristic Curve (AUROC) [68] and the Area Under the Precision-Recall Curve (AUPR) [69], with higher values indicating more effective differentiation between pair types. To illustrate the effectiveness of our I-Div algorithm, we compare it with a selection of representative contrastive algorithms, each adapted to our specific task. All approaches quantitatively evaluate hypothesis applicability by measuring the distribution discrepancy between training and test distributions. These algorithms, which include Maximum over Softmax Probability (MSP) [59], Non-Negative Bregman Divergence (NNBD) [58], Maximum Mean Discrepancy with Deep kernels (MMD-D) [53], and R-Div [47], assess hypothesis applicability through the lens of distribution discrepancy.

### D.2 Additional experiments on semantically dissimilar data

Table 6: Distribution discrepancy of different classes in SVHN. The larger the values of AUROC and AUPR, the better the performance.

| Dataset | Target | MSP | | NNBD | | MMD-D | | R-Div | | I-Div | |
|---------|--------|-----|-----|------|-----|-------|-----|-------|-----|-------|-----|
| | | AUROC | AUPR | AUROC | AUPR | AUROC | AUPR | AUROC | AUPR | AUROC | AUPR |
| | Digit 0 | 100.0 | 100.0 | 98.0 | 98.2 | 83.1 | 84.1 | 100.0 | 100.0 | 100.0 | 100.0 |
| | Digit 1 | 100.0 | 100.0 | 99.8 | 99.8 | 79.3 | 81.7 | 100.0 | 100.0 | 100.0 | 100.0 |
| | Digit 2 | 100.0 | 100.0 | 98.0 | 98.3 | 70.0 | 71.3 | 100.0 | 100.0 | 100.0 | 100.0 |
| | Digit 3 | 100.0 | 100.0 | 97.3 | 97.6 | 69.1 | 71.0 | 100.0 | 100.0 | 100.0 | 100.0 |
| SVHN | Digit 4 | 100.0 | 100.0 | 92.3 | 92.4 | 71.2 | 73.0 | 100.0 | 100.0 | 100.0 | 100.0 |
| | Digit 5 | 100.0 | 100.0 | 96.2 | 96.6 | 80.4 | 81.8 | 100.0 | 100.0 | 100.0 | 100.0 |
| | Digit 6 | 100.0 | 100.0 | 96.5 | 96.8 | 80.7 | 82.5 | 100.0 | 100.0 | 100.0 | 100.0 |
| | Digit 7 | 100.0 | 100.0 | 90.2 | 89.8 | 75.2 | 77.1 | 100.0 | 100.0 | 100.0 | 100.0 |
| | Digit 8 | 100.0 | 100.0 | 96.5 | 97.0 | 78.1 | 80.0 | 100.0 | 100.0 | 100.0 | 100.0 |
| | Digit 9 | 100.0 | 100.0 | 97.8 | 98.0 | 75.5 | 77.4 | 100.0 | 100.0 | 100.0 | 100.0 |

### D.3 Additional experiments on corrupted data

Fig. 1(a) shows a steady decline in the classification performance of the standard network in the test dataset as the noise rate increases. This decline results from the loss of class label-relevant information in the samples as noises increase, hindering the ability of the hypothesis to predict class labels accurately. The expectation is that the distribution discrepancy between the training and test distributions will increase with added noises, suggesting a decrease of hypothesis applicability from the clean training dataset to the corrupted test dataset without accessing the ground truth labels of test samples. However, the results in Fig. 1(b), Fig. 1(c) and Fig. 1(d) show that the proposed I-Div algorithm performs well in this respect. As the classification accuracy of the standard network in the test samples decreases, the ability of I-Div to discriminate between the two datasets increases. While Fig. 1(c) shows the improving ability of HDR to distinguish the two datasets with an increasing noise

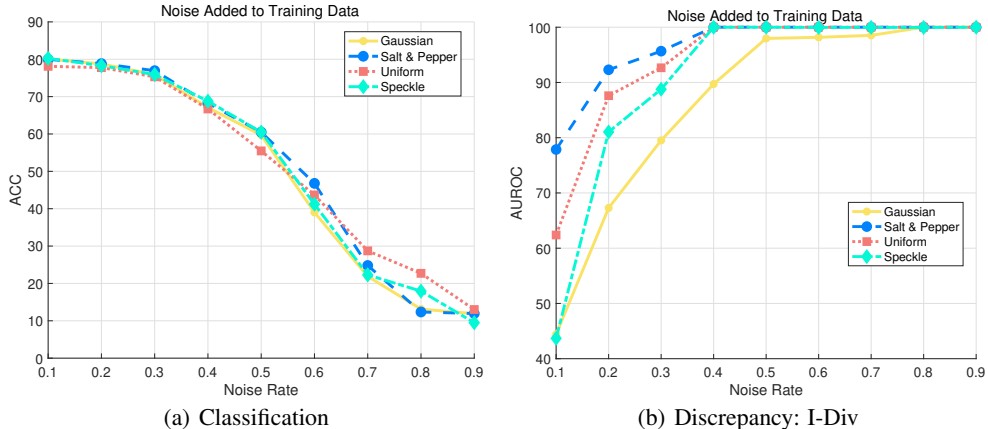

<p style="text-align:center">(a) Classification        (b) Discrepancy: I-Div</p>

Figure 4: Distribution discrepancy between original data and its corrupted variants with different noise rate. (a) shows the classification performance of the standard network for the test datasets containing corrupted samples. (b)(c)(d) present the distribution discrepancy in terms of AUROC.

rate, its performance is not as consistent as that of I-Div, whose discrimination power more closely aligns with changes in the classification ability of the standard network. Fig. 1(d) indicates that R-Div is extremely sensitive to data variation, achieving a complete separation of the two datasets at any noise level. R-Div can differentiate between clean and corrupted datasets even with minimal noises, but this does not indicate whether the hypothesis derived from training dataset is applicable to the test dataset.

The comparison between Fig. 1(b) and Fig. 1(d) can be seen as an ablation study, since HDR is a component of I-Div. From this analysis, it is clear that I-Div outperforms HDR in determining the hypothesis applicability in the test dataset more accurately. HDR relies only on input data for estimating the input density ratio, a metric unrelated to specific model performance. In contrast, I-Div uses this density ratio to estimate the disparity of hypothesis performance between two datasets, offering a more accurate reflection of hypothesis applicability in the test dataset.

### D.4 Additional experiments on adversarial data

As shown in Fig. 2(a), the classification accuracy of a standard network on adversarial samples markedly diminishes with increasing perturbation magnitude. However, as depicted in Fig. 2(d), I-Div maintains a consistently low AUROC, indicating its inability to differentiate between the training and test datasets, akin to human visual perception. This suggests that, even when the classification performance of the standard network is compromised, I-Div still perceives the knowledge from training dataset as transferable to the test dataset. This finding underscores the need to focus on improving the generalization of a standard network to adversarial samples, corroborating our empirical understanding and existing knowledge in the field. Fig. 2(b) presents the AUROC of HDR, which remains high across various perturbation magnitudes, indicating its effectiveness in distinguishing original and adversarial samples. This observation might suggest that, while HDR is responsive to distribution changes induced by adversarial perturbations, it may not accurately reflect the hypothesis applicability from the training to the test datasets. Fig. 2(c) shows the AUROC of R-Div, exhibiting a distinct pattern where the AUROC is relatively lower at the smallest perturbation but improves with larger perturbation. This indicates that R-Div could potentially be used to assess the hypothesis applicability to the test dataset. Nonetheless, in the context of adversarial samples, the results of I-Div imply that enhancing the hypothesis generalization to adversarial samples should be a priority. This conclusion aligns more closely with our intuitive understanding and is supported by numerous advanced research efforts in this area.

### D.5 Additional experiments with different sample sizes and network architectures

We explore the influence of varying sample sizes, denoted as $M$, on the experimental results. This investigation is conducted under the same experimental setup as the one used for the semantically

similar datasets. The findings illustrated in Fig. 3 indicate that for test datasets like CIFAR10.1 and STL10, I-Div consistently exhibits a relatively low AUROC. This pattern suggests that I-Div considers the hypothesis derived from the training dataset to be applicable to these test datasets, a conclusion further supported by the classification results in Table 3. In contrast, for other test datasets with class labels semantically dissimilar to the training dataset, the knowledge learned by the model is deemed non-transferable. As shown in Fig. 3, the ability of I-Div to distinguish these datasets, as quantified by AUROC and AUPR, improves with increasing sample size, eventually reaching a plateau.

Given the hypothesis-oriented nature of the I-Div algorithm, which bases its analysis of hypothesis applicability from training to test datasets, it is pertinent to investigate the impact of varying network architectures on the performance. The findings of these experiments are summarized in Table 5. The data reveal that, across all tested hypotheses, there are consistently low AUROC and AUPR scores for test samples with semantic similarities to the training datasets, such as CIFAR10.1 and STL10. However, a notable variance of performance is observed when examining test datasets with distinct semantic differences, like SVHN and DTD. This highlights the significant role of network architecture in determining a hypothesis capability to generalize and apply learned knowledge to novel, unseen dataset. The robust and consistent performance of I-Div across diverse network architectures further attests to the algorithm stability and broad applicability.

