# OpenReview forum: "Revealing Distribution Discrepancy by Sampling Transfer in Unlabeled Data"
_NeurIPS.cc/2024/Conference — NeurIPS 2024 poster_

### Official Review · Reviewer_f37S · 2024-06-21

**Soundness:** 3
**Presentation:** 3
**Contribution:** 4
**Rating:** 6
**Confidence:** 2

**Summary:**

The paper proposes a novel concept called Importance Divergence (I-Div) to address the lack of labeling of test samples and to measure the difference between training and testing distributions. Through the computation of importance samples, density ratios, and likelihood ratios, I-Div is able to assess the applicability of hypotheses on different datasets without the need for test labels. The main contribution of this method is that it provides a new way to quantify and reduce the expected risk difference between training and testing distributions, thus enhancing the generalization capability of machine learning models in the face of unknown test data.

**Strengths:**

The study proposes a novel concept, which helps to estimate the difference between training and testing distributions, and adapts to the assessment of generalization ability between different datasets. This method can be tested for applicability on different datasets and task types, and can be widely used in a variety of machine learning scenarios, especially in practical applications when the training and testing data distributions are inconsistent.

**Weaknesses:**

Since this method relies on accurate estimation of the density ratio and likelihood ratio, this may limit the general feasibility of the method in practical applications.

**Questions:**

It is recommended that in future research, more simplified or optimized algorithmic processes be explored to reduce the computational cost and improve the practical feasibility of the method.

**Limitations:**

yes

---

> ### Author Rebuttal · Authors · 2024-08-05
>
> We appreciate your valuable feedback and suggestions on the limitations and potential improvements of our method. Here are our responses.
>
> **W1. General Feasibility**
>
> The estimation of the density ratio and likelihood ratio does not limit the applicability of our method. We transform the issue of unknown test sample labels into a problem of estimating these ratios, which are both straightforward to estimate. The density ratio can be estimated using a lightweight adaptor and Eq. (13) within any given hypothesis. The likelihood ratio can be derived analytically by minimizing the upper bound of the generalization error, as shown in Eq. (17), or it can be simplified to a constant value, along with the following experiments.
>
> Specifically, in practical applications, we can simplify the algorithm using only the density ratio and setting the likelihood ratio to a constant value. As discussed in Line 188 of the paper, setting the likelihood ratio to a constant (e.g., 1) is equivalent to assuming a covariate shift in the data. In contrast, using the adaptive likelihood ratio estimation in Eq. (17) assumes a stronger semantic shift.
>
> To clarify, we add corresponding experiments on comparing the performance before and after simplifying the likelihood ratio, and test them using ResNet18 trained on CIFAR10, with test datasets including SVHN, SEMEION, CIFAR10.1, CIFAR10 corrupted by Gaussian noise and adversarial samples (perturbation factor 0.01). The results show little difference in performance. For instance, both setups clearly distinguish between SVHN and CIFAR10. However, with the more accurate likelihood ratio estimation from Eq. (17), I-Div aligns more closely with human prior knowledge, such as recognizing smaller differences between CIFAR10, CIFAR10.1, and adversarial samples. These results demonstrate that even with a simplified likelihood ratio, our method remains effective and practical for various applications. The density ratio and likelihood ratio estimations are both robust and easy to implement, ensuring the general feasibility of our method in real-world scenarios.
>
> | Method| SVHN| SEMEION| CIFAR10.1| CIFAR10 corrupted by Gaussian noise |Adversarial(0.01)|
> |------------|-------|-------|-------|-------|-------|
> |I-Div(constant)| 100.0 |100.0|43.4|46.5 |28.6|
> |I-Div(adaptive)|100.0|100.0|48.6| 49.2 |38.7|
>
> **Q1. Optimization Process**
>
> Your suggestion aligns well with our current direction, and we indeed have been considering similar approaches.
>
> The current algorithm addresses the challenge of unknown class labels in the test samples using density and likelihood ratios for sampling transfer, as detailed in Eq. (4). This approach makes computational effort on estimating these ratios using an adaptor. Our preliminary idea is to treat both training and test samples as originating from an unknown labeling hypothesis. While we cannot know this labeling hypothesis, we do know the ground truth labels for the training data. We can explore the hypothesis space where this labeling hypothesis resides to derive an upper bound for Eq. (4). By applying the Legendre-Fenchel duality, we can simplify this upper bound and then minimize it to estimate the discrepancy between the two distributions. We would be happy to discuss any suggestions or thoughts you might have on this approach.
>
> We appreciate the constructive suggestions and are committed to incorporating these insights into our future research to improve the applicability and efficiency. Thank you for your thoughtful review and guidance.

---

### Official Review · Reviewer_fpGs · 2024-07-03

**Soundness:** 2
**Presentation:** 2
**Contribution:** 2
**Rating:** 4
**Confidence:** 4

**Summary:**

This paper investigates the applicability of hypotheses derived from training datasets to distinctly different test datasets through a series of experiments using various datasets, including CIFAR10, SVHN, PACS, and Office-Home. T

**Strengths:**

New Algorithm Introduction: The paper introduces the I-Div algorithm, which demonstrates a novel approach to evaluating distribution discrepancies between training and test datasets. This innovation contributes to the understanding of hypothesis applicability in machine learning.

Diverse Experimental Scenarios: The study explores various scenarios, including semantically dissimilar data, semantically similar data, corrupted data, and adversarial data. This comprehensive approach offers a fresh perspective on the generalization and robustness of machine learning algorithms.

Robustness and Generalization: The paper’s exploration of the algorithm’s robustness against noise and adversarial attacks, as well as its performance across different sample sizes and network architectures, adds depth to the quality of the research.

**Weaknesses:**

Outdated Baselines: The baseline comparison, particularly using MMD-D from 2020, is quite outdated. This diminishes the paper's novelty and effectiveness compared to state-of-the-art methods. To enhance the paper's relevance and credibility, it would be beneficial to include more recent benchmark algorithms, preferably those introduced after 2021, or popular models like ViT (Vision Transformer) and CLIP (Contrastive Language-Image Pre-training), which have shown significant impact in computer vision.


Backbone and Feature Encoder: Indeed, the choice of backbone in the paper does not align well with recent developments in computer vision. Modern visual tasks increasingly rely on Transformer-based architectures such as ViT and CLIP, which excel in handling interactions between visual and language modalities. Introducing these modern feature encoders in the experiments would allow the algorithm to be tested on more complex and diverse datasets, better reflecting its generalization ability and practical utility.


Performance of I-DIV in Table 1: I concern about I-DIV achieving 100% performance in Table 1 is valid. In practical applications, achieving 100% accuracy is nearly impossible, especially on complex datasets and real-world scenarios. This might suggest some idealization or overfitting in the experimental setup, particularly in how the model or algorithm is tailored to specific datasets or test conditions. Therefore, it's advisable to provide a more detailed explanation of these experimental results in the paper, discussing their applicability and generalization capability in real-world environments.

**Questions:**

see weaknesses

**Limitations:**

see weaknesses

---

> ### Author Rebuttal · Authors · 2024-08-05
>
> We appreciate your valuable feedback and constructive criticisms. Here are our responses to your identified questions.
>
> **Q1&2. Baselines and Backbones**
>
> The paper has already considered post-2021 algorithms, such as NNBD (ICML’21) [R1] and R-Div (NeurIPS’23) [R2] shown in Tables 1, 2, 4 and 5. To further address your concerns, we add H-Div (ICLR’22) [R3] and utilized updated network architectures (ResNet50, ViT, and CLIP) on diverse datasets (ImageNet, OID, Places365, StanfordCars, Caltech101, and DTD) to revise the paper. The additional experimental results are shown below.
>
> It is important to note that the training data used for the pre-trained CLIP model is not accessible. According to Figure 4 in the CLIP paper [R4], CLIP achieves strong zero-shot performance on ImageNet. We, therefore, consider this dataset as part of the training data to estimate the differences between this dataset and the other test datasets.
>
> The experimental results show that the AUROC values of I-Div better reflect the semantic similarity between datasets and ImageNet, aligning with human intuition. Specifically, ImageNet is more similar to OID and Places365 due to their shared everyday objects and scenes, leading to lower AUROC values of I-Div. Conversely, ImageNet is quite different from Caltech101 and DTD, which focus on specific object categories and textures, resulting in higher AUROC values of I-Div. However, the other algorithms show higher AUROC values across most datasets, indicating they struggle to capture the nuanced semantic and visual similarities between ImageNet and more diverse datasets. This results in less accurate distinctions compared to the I-Div algorithm, which aligns better with human intuition.
>
> | Backbone| Method | OID  | Places365 | StandfordCars | Caltech101 | DTD  |
> |----------|--------|------|-----------|---------------|------------|------|
> | ResNet50 | MSP | 100.0| 100.0| 100.0 | 100.0 | 100.0|
> || NNBD   | 91.7 | 90.5  | 96.6 | 95.3 | 98.7 |
> ||MMD-D  | 94.6 | 93.2| 98.6| 96.5| 100.0|
> ||H-Div  | 100.0| 100.0| 100.0| 100.0| 100.0|
> ||R-Div  | 94.6 | 100.0| 100.0| 100.0| 100.0|
> ||I-Div  | **69.3**| **78.9** | **86.5**| **100.0**| **100.0**|
> | ViT      | MSP | 100.0| 100.0 | 100.0  | 100.0| 100.0|
> ||NNBD   | 88.6 | 98.3 | 95.5 | 96.3| 99.7 |
> ||MMD-D  | 92.6 | 100.0  | 96.3  | 95.6| 100.0|
> ||H-Div  | 100.0| 100.0| 100.0| 100.0 | 100.0|
> ||R-Div  | 92.6 | 100.0| 100.0| 100.0| 100.0|
> ||I-Div  | **62.6**| **74.5** | **79.6**| **100.0**| **100.0**|
> | CLIP     | MSP | 100.0| 100.0| 100.0| 100.0| 100.0|
> ||NNBD   | 93.2 | 91.6| 91.3| 95.9| 100.0|
> ||MMD-D  | 96.3 | 93.7| 94.5| 98.6| 100.0|
> ||H-Div  | 100.0| 100.0| 100.0| 100.0| 100.0|
> ||R-Div  | 91.2 | 100.0| 100.0| 100.0| 100.0|
> ||I-Div  | **61.3**| **71.8** | **83.5**| **100.0**| **100.0**|
>
> **Q3. Performance of I-DIV**
>
> It appears there may have been a misunderstanding regarding the objective of our algorithm. Our goal is for the distribution discrepancy to align with human prior knowledge. As demonstrated in Tables 1 and 2, the effectiveness of the algorithm should be evaluated within the context of specific data. Achieving 100% distinction is not always feasible and does not necessarily indicate superior performance.
>
> In cases where I-Div achieves 100% distinction, it suggests that, according to a given pre-trained network, the training and test data are certainly not from the same distribution. While we agree that achieving 100% classification accuracy in general is unrealistic, if the goal is to simply determine whether data samples come from the same distribution and have clear semantic differences, then achieving 100% distinction can be straightforward.
>
> Specifically, in Table 1, the datasets show clear semantic differences, making it normal and straightforward for algorithms to achieve 100% distinction. This result is not due to model overfitting but rather reflects the inherent separability of the data categories. When datasets have distinct semantic boundaries, algorithms can easily detect these differences, especially with large dataset sizes that provide ample data for learning and differentiation. This phenomenon is consistent with the findings in the literature [R1-R3], where similar distinctions were observed, demonstrating that such results are typical in cases of clearly defined semantic classes. The clarity and volume of the data naturally lead to high accuracy in distinguishing between categories, confirming that the observed distinctions are a standard outcome under these conditions.
>
> Conversely, in Table 2, which involves semantically similar datasets, I-Div achieves around 50% instead of 100% accuracy. This indicates that I-Div struggles to distinguish these samples, which is more intuitive. For example, the PACS dataset includes four domains with different styles, but consistent class semantics. Other algorithms like R-Div and MSP can distinctly separate all datasets, possibly because they do not rely on a pre-trained network and continually seek out differences during training. However, I-Div measures the discrepancy between two datasets based on a pre-trained network, considering the network predictive ability on the test data. If the network performs well on the test data, I-Div finds it challenging to distinguish between training and test data, aligning with our intended outcome.
>
> [R1] Kato et al., Non-negative bregman divergence minimization for deep direct density ratio estimation, ICML, 2021.
>
> [R2] Zhao et al, R-divergence for estimating model-oriented distribution discrepancy, NeurIPS, 2023.
>
> [R3] Zhao et al., Comparing distributions by measuring differences that affect decision making, ICLR, 2022.
>
> [R4] Radford et al., Learning transferable visual models from natural language supervision, ICML, 2021.

---

### Official Review · Reviewer_1gtU · 2024-07-09

**Soundness:** 3
**Presentation:** 3
**Contribution:** 3
**Rating:** 6
**Confidence:** 3

**Summary:**

This paper presents a novel approach called Importance Divergence (I-Div) to address the challenge of measuring the discrepancy between training and test distributions when test sample class labels are unavailable. I-Div transfers sampling patterns from the test distribution to the training distribution by estimating density and likelihood ratios. The density ratio is obtained by minimizing the Kullback-Leibler divergence, while the likelihood ratio is adjusted to reduce the generalization error. Experimentally, I-Div accurately quantifies distribution discrepancy across a wide range of complex data scenarios and tasks.

**Strengths:**

1. **Novel Approach**: I-Div offers a novel method to evaluate distribution discrepancies between training and test data without needing test sample class labels, addressing a significant challenge existing methods struggle with.

2. **Enhanced Generalization Capability**: By transferring sampling patterns and adjusting ratios, I-Div enhances the model's generalization capability, which is particularly beneficial for applications such as out-of-distribution (OOD) sample detection.

3. **Validation**: The paper demonstrates the effectiveness of I-Div through various datasets and scenarios, showing that the proposed method works well across different conditions.

**Weaknesses:**

1. **Limitations of Likelihood Ratio Estimation**: As the authors mention, accurately estimating the likelihood ratio is challenging since test sample class labels are unavailable. The proposed adaptive adjustment only partially mitigates this issue and may not fully resolve it.

2. **Computational Complexity**: Estimating the density and likelihood ratios can be computationally expensive and potentially impractical for large datasets or complex models.

3. **Limited Scope of Experiments**: The experiments still focus primarily on simple datasets. Additional validation on other types of datasets is necessary to confirm the generalizability of I-Div.

**Questions:**

This paper presents a methodological advancement in evaluating distribution discrepancies without test sample class labels. However, the fundamental challenge of accurately estimating the likelihood ratio remains unresolved, potentially compromising the method's accuracy. Additionally, the computational complexity may hinder practical application, especially with large or complex datasets.

1. Since the Likelihood Ratio significantly influences Importance Sampling, it greatly affects the quality of the samples. Consequently, this often results in high variance in the outcomes. I am curious about how the author addressed this stability issue during the implementation.
For example, Fig. 3 illustrates the effect of sample size but does not explain which factor is crucial to controlling the quality of the samples. And the size > 1000 means it requires high computational costs. So please elaborate on some more challenges on this matter.

2. What challenges might be faced in extreme cases, such as when the Adaptive Likelihood Ratio is large? While proving convergence is important, discussing practical implementation difficulties, such as when the test dataset size is relatively bigger or when many classes exist, would improve this paper.

**Limitations:**

I believe this paper does not have any negative societal impact.

---

> ### Author Rebuttal · Authors · 2024-08-05
>
> We appreciate your detailed feedback and thoughtful questions. Here are our responses to your raised concerns.
>
> **W1. Likelihood Ratio**
>
> It is impossible to completely solve this problem because the class labels of the test samples are unknown. However, we theoretically decompose this problem to identify solvable aspects and necessary assumptions. The density ratio can be estimated using Eq. (13) with a lightweight adapter by freezing the pre-trained network, while the likelihood ratio can be derived analytically under the weak assumption of minimizing the generalization error, as shown in Eq. (17). We conduct additional experiments on more complex data (ImageNet, OID, DTD, etc.) and networks (ResNet50, ViT, and CLIP) to demonstrate the effectiveness. Please refer to Table R1 in the PDF within the global response, or the table in our response to Reviewer fpGs.
>
> We don't even need to estimate very accurately. Under a stronger covariate shift assumption, where all sample likelihood ratios are set to 1, reasonable results can still be achieved in practice, as shown in the experiments on CIFAR10 below.
>
> | Method| SVHN| CIFAR10.1| Adver.|
> |------------|-------|-------|-------|
> |I-Div(1)|100.0|43.4|28.6|
> |I-Div(Eq. (17))|100.0|48.6|38.7|
>
> **W2. Computational Complexity**
>
> Estimating the density and likelihood ratios is actually very fast. This is because the density ratio can be estimated using a lightweight adapter, and the likelihood ratio can be quickly derived analytically. We add the following throughput experiments on large data and complex models to validate this.
>
> For the density ratio in Eq. (9), the primary training time is spent on the adaptor for a given pre-trained network. The main bottleneck is the feature dimensions, which is the input size of the adaptor and the output size of the pre-trained network. The results below show the throughput for different dimensions across various pre-trained network architectures. ResNet18 has an input dimension of 28x28, while ViT-B/16 has 224x224. This demonstrates the efficiency for large datasets and complex models, as only the downstream adaptor is trained while the pre-trained network remains frozen. For the likelihood ratio, we can directly obtain its analytical solution using Eq. (17), making its time complexity minimal.
>
> | Dimension|512|1024|2048|
> |------------|-------|-------|-------|
> | ResNet18 | 276.14 | 142.97 | 79.02 |
> | ViT-B/16 | 243.62 | 136.70 | 68.04 |
>
> **W3. Limited Scope**
>
> To address your concern, we add ResNet50 and ViT trained on ImageNet and CLIP to estimate the differences between this dataset and OID, Places365, StandfordCars, Caltech101, and DTD. For the complete results, please refer to Table R1 in the PDF within the global response. A portion of the experiments with ResNet50 is shown below. The outcomes align with human intuition.
>
> | Model| OID|Places365|StandfordCars|Caltech101|DTD|
> |------------|-------|-------|-------|-------|-------|
> |MSP|100.0|100.0|100.0|100.0|100.0|
> | NNBD|91.7|90.5|96.6|95.3|98.7|
> |R-Div|94.6|100.0|100.0|100.0|100.0|
> |I-Div|**69.3**|**78.9**|**86.5**|**100.0**|**100.0**|
>
> **Q1. Sample Quality**
>
> I-Div is highly stable and fast, and the experimental results in the paper are averaged over 100 runs. The experimental variance on CIFAR10 is shown in the table below. The lightweight adapter used to estimate the density ratio requires only a small number of samples, and the subsequent likelihood ratio can be directly derived analytically according to Eq. (17). Additionally, as shown in Figure 3, processing $M=1000$ samples on a single GPU takes only 7.09 seconds.
>
> According to the experimental results in the table, if there is a significant difference between the training and test data distributions, the algorithm can easily identify the differences and converge quickly. When the test data is semantically similar to the training data, the stability is lower, but it improves rapidly as the number of samples increases. For more discussion on efficiency, please refer to our response to your W2.
>
> | Test| 100 | 500|1000|
> |------------|-------|-------|-------|
> |SVHN|99.8±0.2 |100.0±0.0|100.0±0.0|
> |CIFAR100|67.6±0.3 |89.1±0.1|93.9±0.0|
> |CIFAR10.1|48.51±0.6|51.81±0.2|43.21±0.0|
>
> **Q2. Extreme Cases**
>
> In fact, our implementation already considers the issue of extreme values. Our core strategy is to ensure that the density ratio remains within a reasonable range by using appropriate activation functions and subsequently applying a proximal algorithm to keep the likelihood ratio within our defined interval. Additionally, we explain that the sample size does not affect the range of the likelihood. We also add experimental explanations in the table below regarding the impact of the number of samples $M$ on data with many class labels.
>
> Specifically, in constructing the adapter (Eq. (9)), we introduce Softplus and GELU to ensure that the density ratio stays within a reasonable range (experimentally within (0.5, 5)). According to Eq. (17), the likelihood ratio depends on the density ratio and can be analytically solved using this formula. In practice, we project the final values to ensure they fall within our predefined interval, which is set to (0.5, 5) in our experiments.
>
> Regarding the large data issue, the $M$ in Eq. (17) cancels out in the denominator of the second term, so the number of samples does not affect the likelihood value. Also, the number of class labels does not impact the likelihood value since the formula does not include them. Semantically different datasets have minimal bias and can be distinguished. However, semantically similar datasets may be misclassified as different distributions, presenting a lower AUROC, indicating that I-Div cannot distinguish between two subsets from CIFAR100.
>
> | Train| Test | 10 | 50 | 100 | 1000 |
> |------------|------------|-------|-------|-------|-------|
> |SVHN|CIFAR100|94.2±2.6 |97.7±1.1|100±0|100±0|
> |CIFAR100|CIFAR100|83.6±3.2|73.6±1.6|62.4±0.4|50±0|

---

> > ### Comment · Reviewer_1gtU · 2024-08-08
> >
> > Thank you very much for the response. I have thoroughly reviewed the answers, and most of my concerns have been well-addressed. Although the scope of the experiments presented in the paper still is somewhat limited, I would like to give more credit to the methodology proposed in the paper. Therefore, I increase the score one step higher.

---

> ### Comment · Reviewer_1gtU · 2024-08-08
>
> Additionally, I suggest the authors make the code used in all experiments publicly available (if accepted).

---

> > ### Author Response · Authors · 2024-08-09
> > **Thank You for Increasing the Score**
> >
> > Thank you very much for your thoughtful response and for taking the time to thoroughly review our answers. We appreciate your acknowledgment of our methodology and your decision to increase the score. Regarding your suggestions:
> >
> > **Code Availability:** We fully agree with the importance of code transparency. If the paper is accepted, we are committed to releasing the code used in all experiments to ensure transparency and facilitate further research in this area.
> >
> > **Scope of Experiments:** We understand your concern about the scope of the experiments. In this revised version, in addition to addressing your concerns with expanded experiments, we have also conducted further experiments on more complex datasets (ImageNet, OID, Places365, StanfordCars, Caltech101, and DTD) and models (ResNet50, ViT, and CLIP) to validate the effectiveness of our algorithm. Detailed results can be found in Figure 1 of the PDF provided in the global response. These new experiments have also been incorporated into the revised version of the paper.
> >
> > If you have any further suggestions or require additional experiments, we are more than happy to incorporate them. Thank you again for your constructive feedback and for your consideration.

---

### Official Review · Reviewer_yuQt · 2024-07-18

**Soundness:** 3
**Presentation:** 3
**Contribution:** 3
**Rating:** 6
**Confidence:** 3

**Summary:**

This paper proposes a discrepancy to measure the difference between two distributions within a common scenario: the labeled training set distribution and the unlabeled test set distribution. This discrepancy arises from the expected risk difference between these two distributions, considering a model pre-trained on the training samples, and involves the estimation of both the density ratio and the likelihood ratio. Experiments were conducted on different types of data splits and data corruptions to demonstrate its effectiveness.

**Strengths:**

1. The scenario considered for this discrepancy is very practical.
2. The definition of discrepancy is intuitive, and the paper is well-written.
3. The experiments demonstrate the robustness of the discrepancy across various scenarios, providing solid evidence of its effectiveness.

**Weaknesses:**

1. My primary concern revolves around the significant impact of the pre-trained classifier $\hat{h}$ on the proposed I-Div. After obtaining the pre-trained model from the training set, the estimated density ratio is predicted by appending a branch to the last layer of this classifier, and the subsequent Adaptive Likelihood Ratio is further estimated based on the predicted $\hat{r}(x)$. Therefore, I am concerned that inaccuracies in the initial estimation by $\hat{h}$ may amplify errors in the final discrepancy. (Although Theorem 3.2 provides a bound, which measures the upper limit of the distance within the same classifier.) I am unsure whether this could limit the practical effectiveness of the proposed I-Div.

2. The compared methods are not presented under consistent settings regarding label utilization and network parameterization. Aligning these settings could enhance clarity for comparison, although direct application to the scenario proposed by the authors might not be feasible.

**Questions:**

1. Could the authors provide a more detailed explanation of how Eq.16 is derived?
2. In Figure 3, what is the sample size of the data in the first column, and why does it perform better than the sample size of 500?
3. Considering my first concern in Cons, what would happen if we were to swap the training set and testing set in the Corrupted Data Experiment, or if the training set itself is inherently challenging?
4. Experiment 4.4 is intriguing; the proposed I-Div appears to be completely unaffected by adversarial samples. Could you provide a more insightful explanation?
5. It would be beneficial to analyze the sensitivity of hyperparameters （e.g. $\sigma$ and $\gamma$）

**Limitations:**

The proposed I-DIV, rooted in expectation risk, bears similarities to existing measures like R-Div. And due to the unknown labels of the test distribution, the authors propose to estimate the density ratio and likelihood ratio, which introduces uncontrollable errors, especially in complex distributions. Thus, I-DIV is affected not only by the estimation error of the likelihood ratio, as noted by the authors, but also the empirical minimizer, which is difficult to mitigate. I believe this limitation somewhat restricts the practical effectiveness of I-DIV.

---

> ### Author Rebuttal · Authors · 2024-08-05
>
> Thank you for constructive suggestions. We appreciate the opportunity to address your concerns.
>
> **W1. Impact of Pre-trained Classifier**
>
> As discussed in Line 24, our goal is to measure the discrepancy between training and test datasets for a given pre-trained classifier, thereby assessing its applicability on the test data. Therefore, our work does not amplify errors. Rather, as shown in Eqs. (5) and (18), we use density and likelihood ratios to indirectly estimate and magnify the discrepancy.
>
> However, different classifiers have varying generalization abilities, which affect their judgment on this matter. For instance, a classifier with strong generalization might consider samples from different domains as coming from the same distribution, while a classifier with weak generalization would not. Therefore, different pre-trained classifiers will result in different density and likelihood ratios, but this is expected. This can help us bypass the issue of unknown test sample class labels while amplifying the sensitivity to the distribution discrepancy.
>
> **W2. Consistent Setting**
>
> We made every effort to ensure consistency in the experimental setup. When conducting comparative experiments, we used the same network structure for all algorithms. Additionally, we provide the performance of the comparative algorithms with and without using class labels, as shown in the table below.
>
> Specifically, in Tables 1, 4, and 5, the different algorithms all use ResNet18 as the backbone, while in Table 2, they use AlexNet. The following experiments on CIFAR10 show that I-Div achieves better performance in both settings. This is mainly because it is based on a pre-trained classifier and combined with the sampling transfer, indirectly estimating the network applicability on test samples. In contrast, the comparative algorithms directly measure divergence without considering the impact of the pre-trained classifier, leading to results that do not align with intuitive expectations.
>
> | Method| CIFAR10.1|STL10|
> |------------|-------|-------|
> |R-Div w/o labels|84.6|94.5|
> |R-Div w/ labels |100.0|100.0|
> |I-Div |**65.3**|**78.9**|
>
> **Q1. Derivation of Eq. (16)**
>
> Theorem 3.2 derives an upper bound on the generalization error for distribution discrepancy w.r.t. the likelihood ratio. To minimize the generalization error bound, we extract the two terms related to the likelihood ratio and use $\gamma$ to balance them. With the constraints revealed by Eq. (15), we then derive Eq. (16).
>
> **Q2. Sample Size**
>
> In Figure 3, the first column represents 100 samples, but the overall performance is worse than that with 500 samples. The values of CIFAR10.1 and STL10 (where lower values are better), which are semantically related to CIFAR10, are similar for 100 and 500 samples and decrease with increasing the sample size. However, for other datasets unrelated to CIFAR10 (where higher values are better), the AUROC and AUPR values are higher with 500 samples.
>
> **Q3. Data Swap and Challenging Data**
>
> This is indeed a very interesting idea. In the revised version, we add experiments with data swapping and challenging datasets. By swapping the data, the network trains on noisy data and tests on clean data. Results show that higher noise levels reduce the ability to distinguish between datasets, aligning with our intuition. For experiments with more challenging data (ImageNet, OID, DTD, etc.) on complex models (ResNet50, ViT, and CLIP), please refer to Table R1 in the PDF within the global response, or the table in our response to Reviewer fpGs.
>
> | Train|0.3|0.5|0.7|
> |------------|-------|-------|-------|
> |Uniform|57.9|53.5|48.6|
> |Salt & Pepper|50.7| 48.7|42.1|
>
> **Q4. Intriguing Experiments**
>
> I-Div aligns more closely with human intuition, as one also struggles to differentiate adversarial samples. I-Div, based on a pre-trained network without adversarial defense, perceives adversarial samples as ``normal" and assigns them with high confidence scores, making it hard to distinguish between original and adversarial samples. Other algorithms detect subtle differences, concluding these samples are from different distributions.
>
> **Q5. Hyperparameter Analysis**
>
> Per your suggestion, in the revised version, we add parameter analysis for $\lambda$ and $\gamma$ using ResNet18 and CIFAR10. The results are shown below. The Lagrange coefficient $\lambda$ stabilizes beyond a certain value, as explained in Appendix B. The algorithm is not very sensitive to $\gamma$, with slightly better results at larger values, consistent with Theorem 3.2. Based on these findings, both parameters are set to 1 by default, as noted in Appendix D1.
>
> | Parameter|Value|SVHN|CIFAR10.1| Adver.|
> |-----------|-------|------|-----------|--------------------|
> |$\lambda$|0.01|67.5|78.9|27.6|
> ||0.1|100.0|44.5|27.9|
> ||1|100.0|43.4|28.6|
> | $\gamma$|0.1|100.0|54.6|28.4|
> ||1|100.0|43.4|28.6|
> ||10|100.0|45.6|29.1|
>
> **L1. Measurement Differences**
>
> As discussed in Appendix A1, I-Div and R-Div are fundamentally different. R-Div measures the difference between training and test datasets using hypotheses from mixed data, requiring known test labels. I-Div measures this difference for a given hypothesis using density and likelihood ratios, even without test labels.
>
> While it is impossible to address unknown test labels without any assumptions, our algorithm easily mitigates this by estimating density and likelihood ratios. In Eq. (13), the density ratio is straightforward to estimate using a lightweight adaptor. The likelihood ratio, derived analytically by minimizing the generalization error bound as shown in Eq. (17), can be estimated with a weak assumption. Even if the likelihood ratio is set to 1, as shown in the CIFAR10 experiments below, our algorithm remains effective.
>
> | Method| SVHN| SEMEION| CIFAR10.1| CIFAR10(Gaussian)|Adver.|
> |------------|-------|-------|-------|-------|-------|
> |I-Div(1)|100.0|100.0|43.4|46.5|28.6|
> |I-Div(Eq. (17))|100.0|100.0|48.6| 49.2|38.7|

---

> > ### Comment · Reviewer_yuQt · 2024-08-12
> >
> > Thank you very much for the response. I still have some questions regarding using noisy labels as the training set and clean labels as the test set. Can we assume that when the noise rate is small, they should belong to the same distribution, but as the noise gradually increases, they should be considered as two different distributions, which is consistent with the results in Figure 1(d). On the other hand, when noisy labels are used as the training set, the classification accuracy will decrease as the noise rate increases, but the distance between the two distributions is also increasing. Therefore, there should be some kind of turning point in the results (I am not sure).
> >
> > And what I want to express in W1 is that whether it's your estimated $\hat{r}(x)$ or $\hat{v}(x, y)$, they both depend on your Pre-trained Classifier. If the Pre-trained Classifier has errors, will these errors be propagated to the estimation of $\hat{r}$ or $\hat{v}$, eventually leading to a significant discrepancy between the measured discrepancy and the actual value?
> >
> > Additionally, regarding Experiment 4.4, if I-Div is robust against adversarial attacks merely because it fails to detect the existence of adversarial samples, can we understand that I-Div appears more robust compared to R-Div simply due to the difference in label settings, rather than the method itself?

---

> ### Author Response · Authors · 2024-08-12
> **Response to Reviewer yuQt**
>
> Thank you for your insightful questions. We have carefully considered your comments and would like to provide the following responses:
>
> **Noisy Data:**
>
> There is indeed a turning point as you suggested. As shown in the table below, this turning point appears at a higher noise rate, such as 0.8.
>
> We previously only presented results for the 0.3 to 0.7 noise range to highlight this interesting finding. In this range, it seems that our algorithm is not very effective at distinguishing between the training and test samples. However, this aligns with our intuition because even with these noise levels, the model still maintains a certain level of generalization to the test data. Specifically, since we added noise to the original data, this can be considered a form of data augmentation, where a certain amount of noise might even enhance generalization. The proposed I-Div relies on a pre-trained classification, which is closely tied to the model generalization ability.
>
> When the noise level is significantly higher, such as at 0.8, the situation changes. At this point, the noise overwhelms the original data, making it difficult for the model to generalize. The noise introduces large variations that no longer match the true data distribution, leading to a clear separation between the training and test sets.
>
> We are truly grateful for you pointing out this interesting aspect. We will continue to delve into the relationship between model generalization and data discrepancy in future research, especially in the context of noisy data.
>
> | Train|0.3|0.5|0.7|0.8|
> |-------|-------|-------|-------|-------|
> |Uniform|57.9|53.5|48.6|100.0|
> |Salt & Pepper|50.7| 48.7|42.1|100.0|
>
> **Error Propagation:**
>
> $\widehat{r}$ and $\widehat{v}$ do indeed rely on the pre-trained classifier. I-Div is designed to measure the difference between training and test data for any given pre-trained network, which in turn helps assess its applicability on test data. This does not involve the type of error propagation typically seen in non-end-to-end training. If I-Div shows a significant discrepancy between two semantically similar datasets when applied to a given pre-trained classifier, this is not due to error propagation. Rather, it indicates the weak generalization ability of the classifier itself, meaning the classifier struggles to generalize well between those two datasets.
>
> To further clarify, you can think of $\widehat{r}$ and $\widehat{v}$ as tools for analyzing the sensitivity of a given network to different data, similar to using a detector to identify out-of-distribution samples for a specific network. Therefore, the nature of the pre-trained classifier, whether it is highly accurate, somewhat flawed, or even based on randomly generated data, does not affect the validity of our approach. I-Div effectively evaluates the model generalization ability regardless of the specific characteristics of the pre-trained classifier. Even with a classifier trained on random data, our method can still measure how well the model generalizes across different datasets.
>
> That said, in our experiments, we used a network trained on the training dataset. While this network generalization ability may not be exceptionally strong, it does possess some degree of generalization, allowing us to measure the differences between its training and test datasets. Naturally, different training methods and network architectures will result in different pre-trained classifiers due to varying levels of generalization ability. Consequently, this will also lead to different data discrepancy measurement results, as demonstrated in our Table 3.
>
> **Adversarial Samples:**
>
> The robustness of I-Div compared to R-Div is not simply due to differences in label settings. The following table highlights that the additional experiments provide further clarification on your point
>
> Regardless of whether labels are present, many current methods, including R-Div, tend to maximize the differences between data (based on our experimental observations). These methods assume that any difference in data implies different distributions. This behavior persists even in labeled settings, where the methods detect even the slightest variations in input data and conclude they belong to different distributions.
>
> However, I-Div does not fall into the trap of detecting these minor differences. Instead, it evaluates data discrepancies based on the model generalization ability. When given adversarial samples, I-Div focuses on the overall information that the model considers for generalization. If the model deems it should generalize to these samples, the perceived difference between the datasets becomes smaller.
>
> | Method |Adversarial(0.01)|
> |------------|-------|
> |R-Div w/o labels |100.0|
> |R-Div w/ labels |100.0|
> |I-Div|38.7|
>
>
> Thank you for your feedback. If you have any more questions or concerns, feel free to reach out.

---

> > ### Author Response · Authors · 2024-08-12
> > **Corrected Results on Noisy Data**
> >
> > **Noisy Data and Q3:**
> >
> > Thank you for your observation. We realized there was a small mistake in our previous experiment, which we have now corrected.
> >
> > As you can see, the corrected results show that when the noise level is low, the model maintains strong generalization ability, making it difficult to distinguish between clean and noisy data. However, as the noise level increases, the model ability to generalize decreases, and the distinction between clean and noisy data becomes more evident. This allows our algorithm to more effectively differentiate between the two distributions.
> >
> > We hope this resolves your concern, and if you have any further questions, please don’t hesitate to ask.
> >
> > |Train|0.1|0.3|0.5|0.7|0.9|
> > |-------|-------|-------|-------|-------|-------|
> > |Uniform|62.3|89.8|100.0|100.0|100.0|
> > |Salt & Pepper|76.9| 100.0|100.0|100.0|100.0|

---

> > > ### Comment · Reviewer_yuQt · 2024-08-12
> > >
> > > I understand your point. What truly impacts the method's performance is actually the classifier's generalization ability. A classifier that performs less well on the training set might somehow perform better with better generalization. This makes sense to me. Therefore, I am willing to raise my score to 6. Additionally, I believe that the discussion about Q3 is important for this method, so please include more detailed experiments and analysis in the revised version.

---

> > > > ### Author Response · Authors · 2024-08-12
> > > > **Thank you for raising the score**
> > > >
> > > > We sincerely appreciate the insightful experimental suggestion you provided, which has led us to explore more intriguing research questions and inspired our ongoing research. We have incorporated the complete experiment in the revised version and will continue to refine the paper.
> > > >
> > > > If you have any further concerns or suggestions, we would be eager to discuss them.

---

### Author Rebuttal · Authors · 2024-08-05

We thank all reviewers for constructive comments and are encouraged by the overall positive feedback from the review. Specifically, the reviewers found that our work addresses an important and practical problem (Reviewers yuQt, 1gtU, fpGs, f37S), introduces a novel and intuitive approach (Reviewers yuQt, 1gtU, fpGs, f37S), and demonstrates robustness and generalization through comprehensive experiments (Reviewers yuQt, 1gtU, fpGs). Additionally, the reviewers appreciated the presentation of the paper (Reviewers yuQt, 1gtU, f37S).

We have made significant changes in creating a new version of this paper by addressing all the comments and substantially improving the paper quality. The main changes include:

1. Added extensive experiments on more complex datasets (ImageNet, Open Images Dataset (OID), Places365, StanfordCars, Caltech101, and DTD), models (ResNet50, ViT, and CLIP), and additional comparison algorithms (H-Div). The experimental table is included in the PDF;

2. Added parameter analysis ($\lambda$ and $\gamma$), time complexity evaluation, and ablation studies (with and without class label information, likelihood ratio estimation based on weak and strong assumptions);

3. Expanded discussions on the limitations, potential solutions, experimental setups, formula derivations, and experimental results, along with the relevant experiments;

4. Included a detailed explanation to the impact of pre-trained classifier and the likelihood ratio derived from the generalization error bound.


We sincerely thank the reviewers for their time and effort in evaluating our work and providing valuable suggestions for improvement. We are committed to continuously enhancing our research and its value to the community. Please feel free to reach out if there are any further questions or clarifications needed. We look forward to presenting our improved work to you.

---

### Decision · Program_Chairs · 2024-09-25

**Decision:**

Accept (poster)

**Comment:**

This paper introduces a discrepancy measure to quantify the difference between two distributions, i.e., the labeled training set distribution and the unlabeled test set distribution. This discrepancy is derived from the expected risk difference between these distributions, built on a model pre-trained on the training data. The experimental results show the general applicability of the proposed method. The reviewers acknowledge the novelty and the effectiveness of the proposed method. I recommend it for acceptance.